# Chronic cough relief by allosteric modulation of P2X3 without taste disturbance

Chang-Run Guo [1,2,3,8], Zhong-Zhe Zhang[1,8], Xing Zhou[1,8], Meng-Yang Sun[1], Tian-Tian Li[1], Yun-Tao Lei[1], Yu-Hao Gao[1], Qing-Quan Li[1], Chen-Xi Yue[1], Yu Gao[1], Yi-Yu Lin[1], Cui-Yun Hao[1], Chang-Zhu Li[4], Peng Cao [5], Michael X. Zhu [6], Ming-Qiang Rong[7] ✉, Wen-Hui Wang [1] ✉ & Ye Yu [1,3] ✉

P2X receptors are cation channels that sense extracellular ATP. Many therapeutic candidates targeting P2X receptors have begun clinical trials or acquired approval for the treatment of refractory chronic cough (RCC) and other disorders. However, the present negative allosteric modulation of P2X receptors is primarily limited to the central pocket or the site below the left flipper domain. Here, we uncover a mechanism of allosteric regulation of P2X3 in the inner pocket of the head domain (IP-HD), and show that the antitussive effects of quercetin and PSFL2915 (our nM-affinity P2X3 inhibitor optimized based on quercetin) on male mice and guinea pigs were achieved by preventing allosteric changes of IP-HD in P2X3. While being therapeutically comparable to the newly licensed P2X3 RCC drug gefapixant, quercetin and PSFL2915 do not have an adverse effect on taste as gefapixant does. Thus, allosteric modulation of P2X3 via IP-HD may be a druggable strategy to alleviate RCC.

P2X receptors are extracellular ATP-activated non-selective cation channels consisting of seven subtypes, P2X1-7, which are found in most tissues as homo- or hetero-trimers. These channels participate in various pathophysiological processes such as immunomodulation, neurotransmission, pain sensation, thrombosis, tumors, and neurological diseases, and are considered as a class of drug targets[1,2]. Several antagonists against P2X receptors have entered clinical trials and shown some therapeutic efficacy, such as the P2X7 antagonists CE-224,5353 and AZD90564 (rheumatoid arthritis)[3], GSK-1482160 (inflammation)[4], AFC-5128 (neuropathic pain, multiple sclerosis, and gastrocnemius dystrophy)[5], JNJ-54175446 (major depressive disorder)[6], JNJ-55308942 (mood disorders)[7], SGM-1019 (nonalcoholic steatohepatitis)[8], BIL010T (melanoma)[9] and RQ-00466479 (neuropathic pain)[10], which are in Phase I or Phase II clinical studies, and the P2X4 antagonist NC-2600 in trials for the treatment of neuropathic pain[11]. Gefapixant (also known as AF-219 or MK-7624) is a P2X3 receptor antagonist for refractory chronic cough (RCC), and the only P2X-targeting drug currently available on the market and the first drug for RCC in more than 60 years since the approval of dextromethorphan in 1958[12,13]. Because dextromethorphan targets opioid receptors, it has been associated with infant mortality and hallucinogenic drug-like adverse effects at high doses[14,15]. Therefore, strategies

[1]School of Basic Medicine and Clinical Pharmacy, China Pharmaceutical University, Nanjing 211198, China. [2]School of Traditional Chinese Pharmacy, China Pharmaceutical University, Nanjing 211198, China. [3]State Key Laboratory of Natural Medicines, China Pharmaceutical University, Nanjing 211198, China. [4]State Key Laboratory of Utilization of Woody Oil Resource, Hunan Academy of Forestry, Changsha, Hunan 410004, China. [5]Hospital of Integrated Traditional Chinese and Western Medicine, Nanjing University of Chinese Medicine, Nanjing 210023, China. [6]Department of Integrative Biology and Pharmacology, McGovern Medical School, The University of Texas Health Science Center at Houston, Houston, Texas 77030, USA. [7]The National & Local Joint Engineering Laboratory of Animal Peptide Drug Development, College of Life Sciences, Hunan Normal University, Changsha, China. [8]These authors contributed equally: Chang-Run Guo, Zhong-Zhe Zhang, Xing Zhou. ✉e-mail: rongmq@hunnu.edu.cn; whwang@cpu.edu.cn; yuye@cpu.edu.cn

are also being sought for the modification of RCC using non-opioids, including corticosteroids, antibiotics[16], and some dietary supplements such as quercetin[17]. The mechanism by which quercetin alleviates cough symptoms is generally believed to result from its antioxidant and anti-inflammatory activities, but whether there are other receptor-specific mechanisms of action is unclear[18–21]. In addition, although gefapixant is approved for marketing in Japan in 2022, it suffers from a significant drug-related adverse effect of taste disturbance[12,22–24]. Newer small molecule drug candidates, such as BAY-1817080 (Bayer)[25,26], BLU-5739 (Bellus)[27], and S-600918 (Shionogi)[28], which were aimed to circumvent this side effect, have slightly weaker clinical efficacy than gefapixant (Supplementary Table 1). Therefore, there is still a need to develop small molecule drugs targeting P2X3 based on new allosteric mechanisms for the treatment of RCC.

Allosteric modulation is considered more druggable than orthosteric modulation based on the enhanced selectivity and reduced adverse effects found in the drug development for G-protein coupled receptors (GPCRs), ion channels, and kinases[29]. The extracellular portion of the P2X receptor/channel contains several structural domains, namely the head, left flipper (LF), dorsal fin (DF), and lower body (LB) domains[30]. Two P2X allosteric modulation sites have been well defined: one is located below the orthosteric ATP-binding site of the human P2X3 (hP2X3) receptor, which we have previously shown to be composed of the LF and LB domains of one subunit, and the DF domain of the adjacent subunit (gefapixant's binding site)[5,31]. The other is in the central pocket of the P2X receptor, such as the binding site of JNJ47965567, A740003, A804598, GW791343, and AZ10606120 for P2X7[32] and that of BX430 for P2X4[33,34]. More allosteric sites have allegedly been discovered, although this information is only supported by in silico docking or other computational approaches without more experimental validations[35–41]. Therefore, one of the essential aspects in accelerating P2X drug development is to identify new druggable P2X allosteric regulatory mechanisms.

The amino acid sequences of the head, DF and LF domains are not conserved across P2X subtypes[42,43], and allosteric modulation acting at these regions would result in highly subtype-selective P2X inhibitors,

such as gefapixant that blocks relative motions of the LF and DF domains[31] with $IC_{50}$ values (the concentration yielding half of inhibition) of ~40–200 nM in human P2X3 (hP2X3) and >10 μM in P2X1, P2X2, P2X4 and P2X7[5,44–46]. The head domain is also a region of sequence variation among P2X subtypes, which is involved not only in the activation but also in the desensitization of the P2X receptor[2,47]. The groove within the subunits between the head domain of one subunit and the DF region of the adjacent subunit of P2X2 is referred to as the ATP-binding site jaw, and movement of the head and DF domains during ATP activation lead to closure of the ATP-binding site jaw and downward movement of the LF region[48]. Whether this mechanism occurs in all P2X subtypes needs further studies. Similarly, most of the small molecules that interact with the central pocket of the P2X receptor are subtype-specific due to the highly variable amino acid compositions of this region[49]. All above allosteric processes provide valuable hints for discovery of new P2X-targeting allosteric drugs; however, identifying novel allosteric mechanism is also crucial for the development of P2X medicines.

Here, we find that P2X3 activation is different from P2X2. Instead of closing the ATP-binding site jaw, a leftward shift of the lower loops of the head domain and tightening of the inner pocket of the head domain (IP-HD) are necessary for P2X3 activation. Based on the allosteric change of IP-HD, we screen the TARGETMOL chemical library to obtain quercetin, and optimize quercetin to obtain PSFL2915, a cough-suppressant molecule with nM affinity for hP2X3. We also demonstrate that quercetin and PSFL2915 inhibit P2X3 activation by interrupting the allosteric change of IP-HD. Notably, quercetin and PSFL2915 have no side effect of taste dysregulation while being therapeutically equivalent to the P2X3-targeting RCC suppressant gefapixant.

## Results

### Tightening of the IP-HD upon binding of ATP to hP2X3

By superimposing the hP2X3 resting (PDB ID: 5SVJ) and open (PDB ID: 5SVK) structures, we found that the ATP-binding site jaw did not undergo tightening upon channel activation (Fig. 1a, b). Although there is some upward movement of the DF region during hP2X3 opening, the

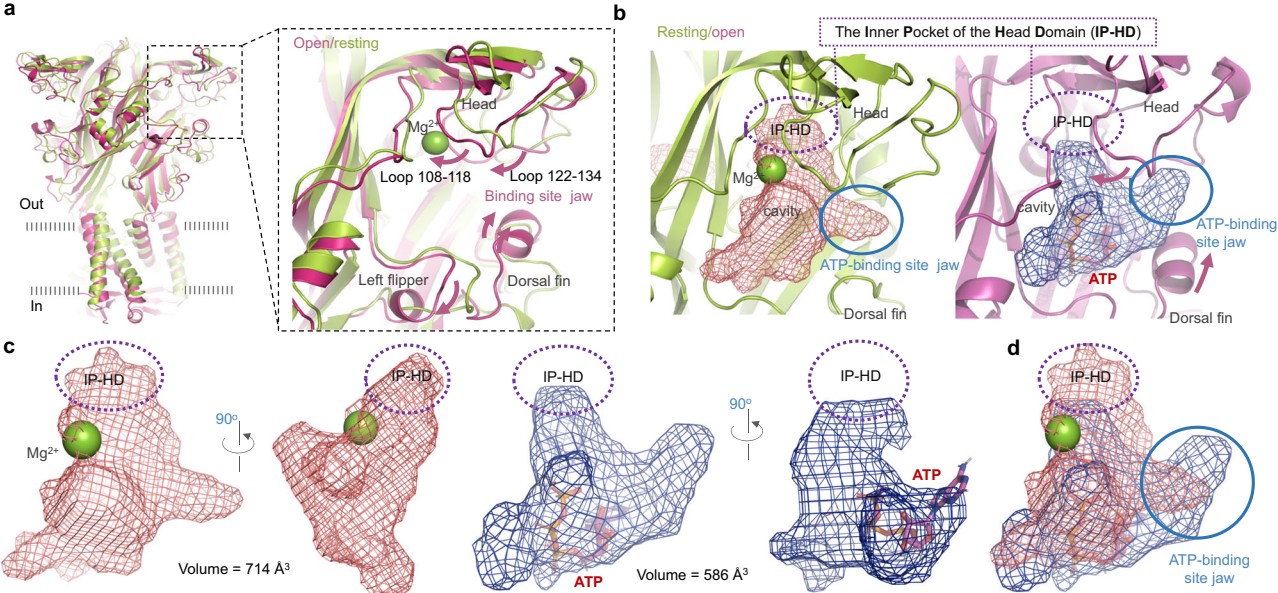

**Fig. 1 | Shrinkage of the internal pocket of the head domain (IP-HD) is one of the major conformational changes of human P2X3 (hP2X3) from the resting to open states. a** Side view and zoomed-in view of superimposed structures of the open (light purple, PDB ID: 5SVK) and resting (green, PDB ID: 5SVJ) states of the human P2X3 (hP2X3) receptor. **b** Side view of the cavity formed by the left flipper (LF), dorsal fin (DF), and head domains in the open (blue) and resting (brick red) states. The IP-HD (purple dashed circles) and ATP-binding site jaw (blue solid circle) are the main changing regions from the resting to the open state. **c** The volumes of the IP-HD in resting (brick red) and open (blue) states. **d** Transition from the resting to the open state. The volume of IP-HD (purple dashed circles), but not the ATP-binding site jaw (blue solid circle), shrinks significantly.

flexible loops on the lower head domain, including loops formed by residues 108–118 and 122–134, shifted to the left to evade the DF domain's upward shifting (Fig. 1a and Supplementary Fig. 1a); as a result, the entire ATP-binding site jaw does not close (Fig. 1b, right panel), which is different from the allostery that has been demonstrated in P2X2[48].

Interestingly, based on the structural comparison of the resting and open states, the volume of IP-HD seems to undergo a significant decrease due to the forced leftward shift of loops 108–118 and 122–134 (Fig. 1b–d and Supplementary Fig. 1a–c). The volume of the entire ATP-binding pocket, of which IP-HD is a fraction, decreased from 714 to 586 Å³, and this size reduction mainly resulted from the reduction of the IP-HD volume (Fig. 1b–d and Supplementary Fig. 1b, c). Molecular dynamics (MD) simulations of the resting and ATP-bound open hP2X3 structures for about 0.3-μs showed that loops 108–118 and 122–134 were tied up due to the presence of ATP, as the root mean square fluctuations (RMSF) of these two loops in the open state were smaller than those of the resting structure (Supplementary Fig. 1d). Moreover, the RMSF values of these two loops in the open state are more correlated with the experimental X-ray B-factor for the open state (Supplementary Fig. 1d, lower panel) rather than that for the resting state (Supplementary Fig. 1d, upper panel). Along with the IP-HD tightening, the rotation of the individual chemical bonds of ATP was also limited to a certain degree of fluctuations throughout 0.3-μs simulations (Supplementary Fig. 1e). These observations indicate that the size reduction of IP-HD likely resulted from ATP-binding rather than a unique transient conformation during the experimental X-ray structure determination.

The zinc bridge H120...Zn$^{2+}$...H213-induced tightening of ATP-binding site jaw (Supplementary Fig. 2d and ref. 48) enhances the ATP-evoked current of rat P2X2 (rP2X2) via the natural histidine pair between the head (H120) and DF (H213) domains (Supplementary Fig. 2a–c). To verify that the conformational changes of ATP-binding site jaws differ between P2X3 and P2X2, we constructed a mutant hP2X3 with an engineered zinc bridge by introducing histidine at the equivalent positions as rP2X2, yielding hP2X3$^{Y114H/T202H}$ (Supplementary Fig. 2a–c). After transfection and expressing in HEK-293 cells, whereas 30 μM Zn$^{2+}$ increased the ATP-evoked rP2X2 current (Supplementary Fig. 2d, e), it failed to do so on either wild-type (WT) hP2X3 or hP2X3$^{Y114H/T202H}$ (Supplementary Fig. 2d, e), suggesting that the head and DF domains and the ATP-binding site jaws behave differently between P2X2 and P2X3. This is further supported by the inhibitory rather than a potentiating effect of Zn$^{2+}$ on hP2X3 with another engineered zinc bridge hP2X3$^{L127H/T202H}$ between the head and DF domains (see below).

## IP-HD allosteric change detection via voltage-clamp fluorometry
To directly observe the conformational changes of IP-HD during P2X3 opening, we incorporated a fluorescent unnatural amino acid (flUAA), ANAP (3-(6-acetylnaphthalen-2-ylamino)-2-aminopropionic acid) into hP2X3, where it is sensitive to changes in the residues around its insertion (Fig. 2a)[50]. Codons for key residues within IP-HD were mutated to TAG individually. By expressing tRNA$^{ANAP-CUA}$/tRNA-synthase and adding ANAP to the medium, the residues were replaced by ANAP one at a time. The engineered hP2X3 also contained a yellow fluorescent protein (YFP) at the C-terminus to aid the identification of transfected cells. Cells expressing hP2X3 with the incorporation of flUAA ANAP showed both YFP and ANAP fluorescence signals (Fig. 2b)[51].

We inserted flUAA ANAP at positions 111, 114, 131, 149, 156, 281, and 297 within the IP-HD to create the following: hP2X3$^{E111-ANAP-YFP}$, hP2X3$^{Y114-ANAP-YFP}$, hP2X3$^{G131-ANAP-YFP}$, hP2X3$^{I149-ANAP-YFP}$, hP2X3$^{E156-ANAP-YFP}$, hP2X3$^{R281-ANAP-YFP}$ and hP2X3$^{L297-ANAP-YFP}$ (Fig. 2c and Supplementary Fig. 1c). Despite the presence of flUAA ANAP at these positions, ATP still caused channel opening (Fig. 2c, lower panel). Using voltage-

clamp fluorometry (VCF), we found that all mutants exhibited significant shifts in the wavelength of peak emission after the administration of ATP (Fig. 2c, d). As a control, cells expressing WT hP2X3-YFP, which did not have ANAP incorporation, only exhibited low emission at wavelengths between 400 and 600 nm, which did not change significantly after ATP administration (Fig. 2c, d). We also chose to incorporate ANAP into V143 (hP2X3$^{V143-ANAP-YFP}$), a site that is not expected to change significantly during activation because this residue is directly exposed to the solution and has few other amino acids nearby (Fig. 2c, upper panel). As expected, the emission peak of hP2X3$^{V143-ANAP-YFP}$ fluctuated between −1 nM and 3 nM (Fig. 2c). Consistently, mutations such as hP2X3$^{E111-ANAP-YFP}$, hP2X3$^{G131-ANAP-YFP}$, and hP2X3$^{R281-ANAP-YFP}$, with hP2X3$^{V143-ANAP-YFP}$ as control, still yielded significant differences in the emission peak of ANAP, indicating that the shift in the emission peak of the flUAA ANAP-incorporated mutants was caused by a conformational change in IP-HD rather than by ATP itself.

The fast-opening and fast-desensitizing characteristics of P2X3 could hamper the detection of ATP-induced change in ANAP fluorescence spectrum by VCF. To overcome this potential problem, we took advantage of the Y37A mutation at the cytoplasmic side of hP2X3, which dramatically delays desensitization and increases the opening duration of this channel[52]. With the introduction of Y37A, the hP2X3$^{Y37A/E156-ANAP-YFP}$ mutant exhibited a more significant shift in the wavelength of ANAP peak emission following ATP administration (Fig. 2c). In addition, ATP administration also caused a substantial shift in ANAP peak emission wavelength in hP2X3$^{Y37A/T134-ANAP-YFP}$ (Fig. 2c, d). The addition of Y37A to hP2X3$^{E111-ANAP-YFP}$, on the other hand, did not alter the extent by which ATP shifted the ANAP peak emission wavelengths of hP2X3$^{Y37A/E111-ANAP-YFP}$ as compared to that of hP2X3$^{E111-ANAP-YFP}$ (Fig. 2c). Together, the VCF experiments revealed that the amino acids inside or near IP-HD, including E111, Y114, T134, I149, E156, R281, and L297, are exposed to changed local environments upon ATP binding and channel activation, implicating conformational changes at this region.

## IP-HD tightening and hP2X3 activation are coupled
To confirm the role of IP-HD tightening in P2X3 activation, we employed a crosslinking approach introducing paired cysteine residues within the IP-HD (hP2X3$^{D158C/E111C}$ and hP2X3$^{G131C/E111C}$) and between the head and DF domains (hP2X3$^{L127C/T202C}$) (Fig. 3a). The disulfide bonds formed between the engineered cysteines were expected to restrain the conformational changes at IP-HD, including the leftward shift of loops in the lower head domain (loops 108–118 and 122–134; Fig.1b) and thereby impair ATP-induced channel gating. We confirmed intersubunit disulfide bond formation in hP2X3$^{L127C/T202C}$ using non-reducing gel analysis of lysates from the transfected cells. In the absence of β-mercaptoethanol (β-Me), hP2X3$^{L127C/T202C}$ displayed a distinct tri-subunit molecular weight band, which was converted to the single-subunit molecular weight band with the treatment of β-Me (Fig. 3b). For WT hP2X3 and the single mutants, L127C and T202C, multiple bands including the one for the single subunit were found in the absence of β-Me (Fig. 3b). For hP2X3$^{D158C/E111C}$ and hP2X3$^{G131C/E111C}$, the non-reducing gel analysis could not be used to validate the disulfide bond formation because they form intrasubunit disulfide bonds.

We then examined ATP-evoked currents in cells expressing WT hP2X3 and its double cysteine mutants. A saturating concentration of ATP (10 μM) was applied three times, with the second application preceded by 0.3% H$_2$O$_2$ and the third one preceded by 10 mM DTT to induce and disrupt disulfide bridges, respectively. For hP2X3$^{G131C/E111C}$ and hP2X3$^{L127C/T202C}$, the ATP-induced currents were markedly decreased by the H$_2$O$_2$ treatment but then recovered after the exposure to DTT (Fig. 3c, d), whereas for WT hP2X3 and hP2X3$^{D158C/E111C}$, neither H$_2$O$_2$ nor DTT significantly affected the ATP-induced current (Fig. 3c, d). These results suggest that H$_2$O$_2$ reduced ATP-evoked

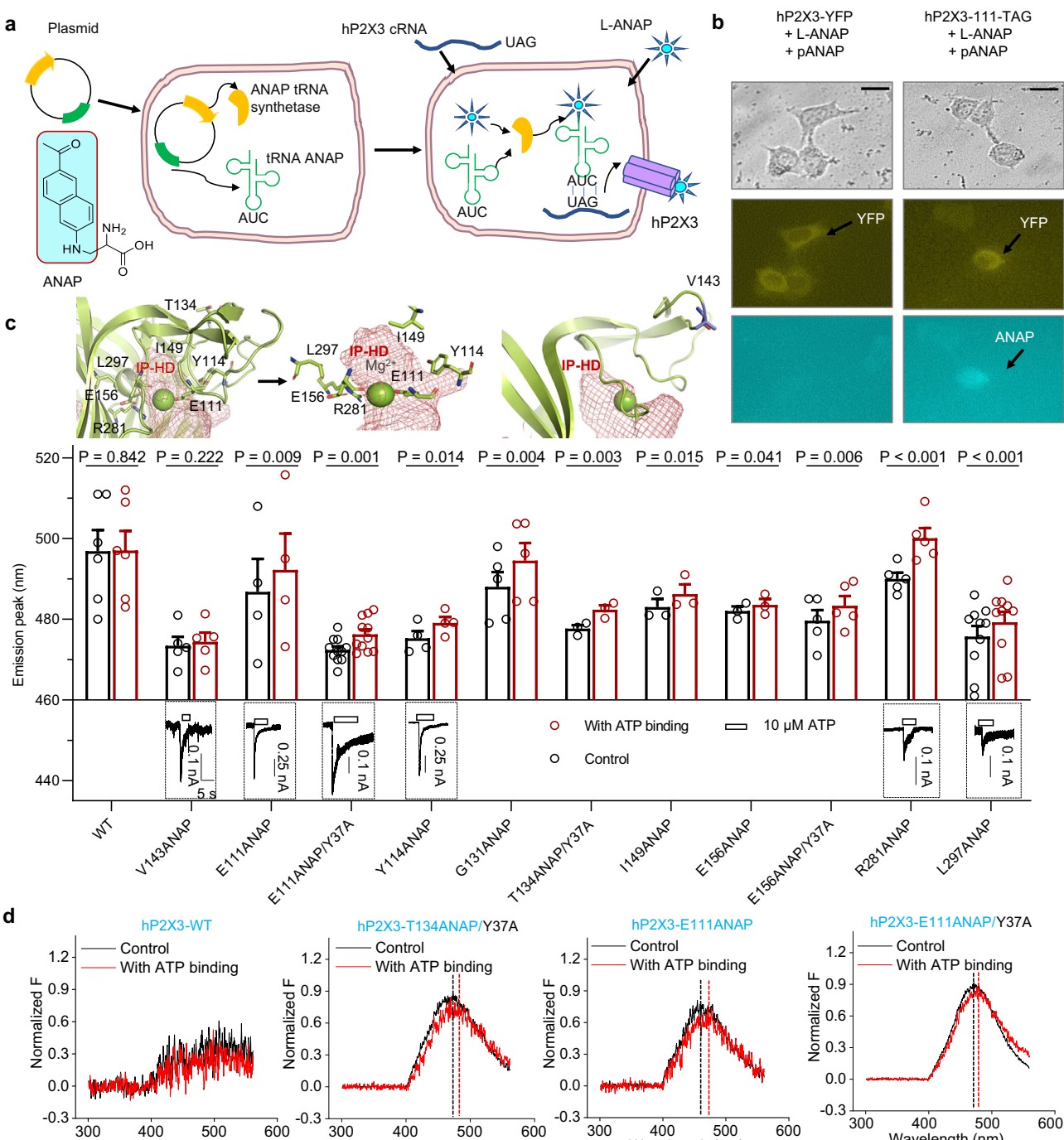

**Fig. 2 | ATP induces conformational changes in IP-HD as revealed by fluorescent unnatural amino acids (flUAA) incorporation and voltage-clamp fluorometry (VCF). a** Chemical structure of ANAP and strategy for incorporating L-ANAP into the P2X3 receptor. Plasmids containing ANAP tRNA synthetase and the corresponding tRNAs, as well as plasmids carrying the genes encoding the receptor, were transiently transfected into HEK293T cells, and then cultured in ANAP-containing medium for at least 24 h to ensure that ANAP was integrated into the proper location of the receptor. **b** Representative images of negative control cells (no pANAP vector added) and positive cells (ANAP-integrated P2X3). Fluorescence of yellow fluorescent protein (YFP) was detected only when cells were transfected with both pANAP and P2X3 receptor plasmids. Pseudo-color was used in ANAP and

YFP (scale bar, 20 μm). The experiment was repeated thrice with similar results. **c** Location of selected sites in the IP-HD of P2X3 and the negative control (upper panel), their responses to saturating ATP (lower panel), and a summary of the shift of the peak of ANAP emission wavelengths after ATP administration (middle), Each circle represents an independent cell; $n = 3$ (T134ANAP/Y37A, I149ANAP, and E156ANAP), 4 (E111ANAP and Y114ANAP), 5 (V143ANAP, G131ANAP, E156ANAP/Y37A, and R281ANAP), 6 (WT), 10 (L297ANAP) or 11 (E111ANAP/Y37A). Data are expressed as mean ± SEM. $P$ value was calculated from a paired, two-tailed $t$ test. **d** Representative emission spectra of cells that expressed WT P2X3 and some of its ANAP-incorporated mutants (black, bath solution; red, solution containing 10 μM ATP). Source data are provided as a Source Data file.

currents in hP2X3$^{G131C/E111C}$ and hP2X3$^{L127C/T202C}$ by promoting disulfide bonds between the engineered cysteine and in turn hampering the conformational change at IP-HD needed for the channel opening, and this effect was reversed by the reducing agent, DTT.

Although $H_2O_2$ and DTT did not significantly alter the ATP-evoked current of hP2X3$^{D158C/E111C}$ (Fig. 3c, d), this mutation diminished the inhibition efficiency of the IP-HD allosteric inhibitor (see below). Most likely, a disulfide bond can form between C158 and C111. However, the

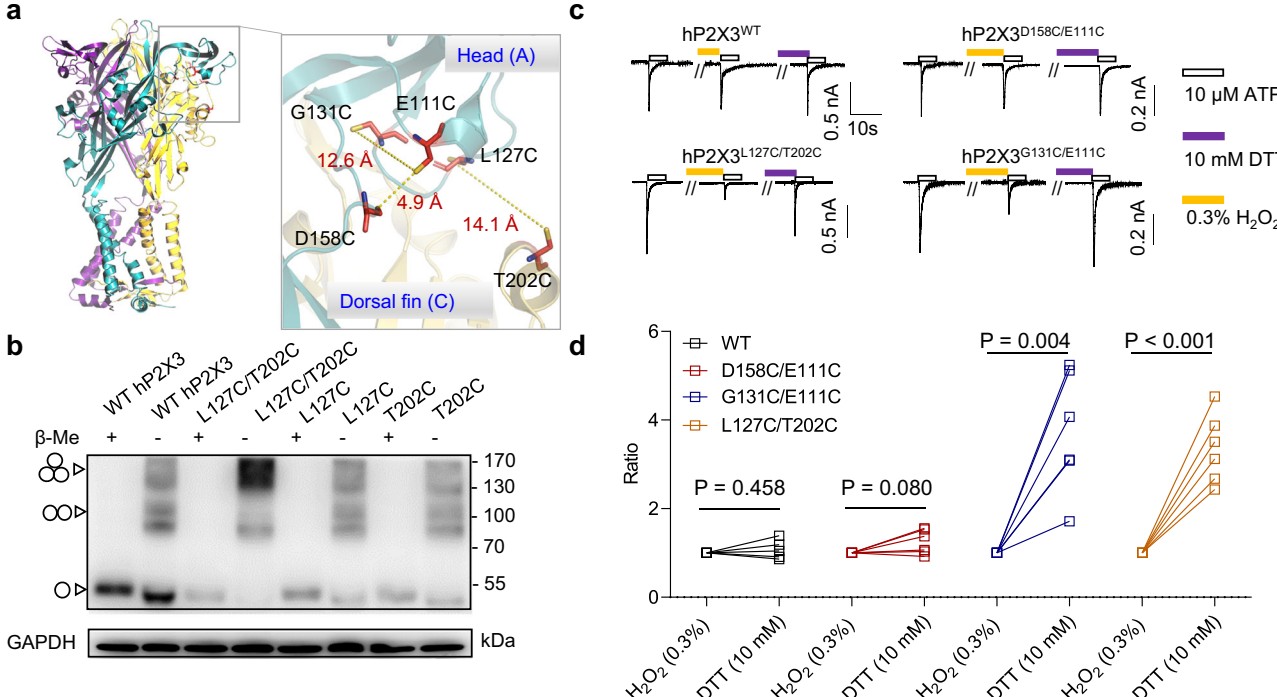

**Fig. 3 | Tightening of IP-HD is essential for ATP-induced P2X3 receptor activation. a** Zoomed-in view of IP-HD. Mutated residues around IP-HD are indicated with sticks for emphasis. **b** Homo-oligomers of hP2X3 with double mutations L127C (in the head domain of one subunit) and T202C (in the DF domain of a neighboring subunit) showed predominantly as trimers in non-reducing Western blots. Cells transfected with WT P2X3 and the cysteine substitution mutants as indicated were lysed in buffers with and without β-ME (1%, 10 mM). Positions corresponding to the sizes of monomeric, dimeric, and trimeric P2X3 subunits are marked with arrowheads, respectively. The experiment was repeated thrice with similar results.

**c**, **d** Representative currents (**c**) and pooled data (**d**) recorded from cells transfected with D158C/E111C ($n = 6$), L127C/T202C ($n = 6$), G131C/E111C ($n = 6$), and wild-type (WT, $n = 5$) hP2X3 receptors. Cells were clamped at −60 mV and currents were induced by ATP (10 μM) at 7 min intervals. $H_2O_2$ (0.3%) and dithiothreitol (DTT, 10 mM) were used to promote and disrupt the disulfide bond, respectively. Data in **d** represent the ratio of ATP-evoked current after to before the DTT treatment. Each line represents an independent cell. One sample two-tailed $t$ test against the value 1. Source data are provided as a Source Data file.

disulfide crosslinking here probably does not significantly affect ATP-induced channel gating because the ~5 Å distance of C158-$C_\alpha$...$C_\alpha$-C111 in hP2X3$^{D158C/E111C}$ with the disulfide bond formation is very similar to the 4.9 Å distance of D158-$C_\alpha$...$C_\alpha$-E111 as determined in the X-ray structure of hP2X3 (Fig. 3a). Thus, the conformational perturbation caused by the disulfide bond was probably insufficient to impair the movement of the IP-HD in response to ATP activation of hP2X3.

Next, we engineered a zinc bridge between loop 122–134 and the DF domain by introducing histidine at positions 127 and 202, i.e., hP2X3$^{L127H/T202H}$ (Fig. 3a). We expected that the bridge formed this way would hinder the ATP binding-induced inward tightening of the loops of the lower head domain (Fig. 1b). Indeed, at both 0.3 and 3 mM, $Zn^{2+}$ significantly attenuated the ATP-induced currents of hP2X3$^{L127H/T202H}$, but not that of WT hP2X3 (ATP-current ratio: $1.33 \pm 0.37$ vs. $0.771 \pm 0.062$, and $0.878 \pm 0.132$ vs. $0.354 \pm 0.012$, hP2X3$^{WT}$ vs. hP2X3$^{L127H/T202H}$ for 0.3 mM and 3 mM $Zn^{2+}$ in standard extracellular solution (SS), respectively; Fig. 4a, b).

Finally, we used covalent occupancy to perturb the conformation of IP-HD to demonstrate that the conformational change of this region is essential for P2X3 activation. We made cysteine mutations, V77C, D158C, I149C, and E156C, in the IP-HD and found that covalent modification of the cysteine with 1 mM 5,5'-dithiobis (2-nitrobenzoic acid) (DTNB) significantly reduced the response to ATP by the mutant channels, but not that by WT hP2X3 (Fig. 4c, d) (ATP-current ratio $_{after/before}$ = $0.523 \pm 0.126$, $0.464 \pm 0.042$, $0.648 \pm 0.071$ and $0.499 \pm 0.071$ for hP2X3$^{V77C}$, hP2X3$^{D158C}$, hP2X3$^{I149C}$ and hP2X3$^{E156C}$, respectively; Fig. 4e). Moreover, the inhibitory effects of DTNB were reversed by 10 mM DTT (Fig. 4c, d), confirming that they were caused by cysteine modification. Together with the VCF experiments showing

the conformational change of IP-HD accompanied with ATP-induced hP2X3 activation, the crosslinking experiments above further demonstrated that the left shift of loops 108–118 and 122–134 and the consequent IP-HD tightening are critical for P2X3 activation.

## Quercetin and PSFL2915 are allosteric inhibitors of hP2X3

Based on the conformation of IP-HD in the resting state (PDB code: 5SVJ), we then performed a virtual screening of the TARGETMOL bioactive compound library (including Natural Compound Library, Library of Already Marketed Drugs, etc., approximately 20,000 compounds; TOPSCIENCE$^{TM}$ (Taoshu), China) and selected 20 natural product hits for verification by electrophysiology (100 μM, Supplementary Fig. 3a). Among the several P2X3 inhibitors confirmed by electrophysiology (Supplementary Fig. 3b, c), we focused on quercetin, a flavonoid found in many fruits and vegetables and sold as a dietary supplement with implications in relieving cough[17,21,53]. Since P2X3 has been clinically validated as a target for the treatment of chronic cough[22,23,54], we characterized the effect of quercetin (Fig. 5a) on P2X receptors to perform a more detailed examination of IP-HD allostery.

First, we examined the selectivity of quercetin among P2X receptor subtypes (except for the two non-functional subtypes P2X5 and P2X6). Quercetin concentration-dependently inhibited hP2X3 and hP2X2, with $IC_{50}$ values of $4.72 \pm 0.35$ and $17.1 \pm 0.9$ μM, respectively (Fig. 5b). At 100 μM, quercetin inhibited hP2X1 by $21.4 \pm 5.0\%$, and caused nearly no inhibition of hP2X4 and hP2X7 (Fig. 5b), suggesting that quercetin is relatively selective among the functioning P2X subtypes. Additionally, quercetin showed no noticeable preference between the homotrimeric hP2X3 ($IC_{50} = 4.72 \pm 0.35$ μM) and heterotrimeric rP2X2/3 receptors ($IC_{50} = 3.64 \pm 0.18$ μM, Fig. 5b).

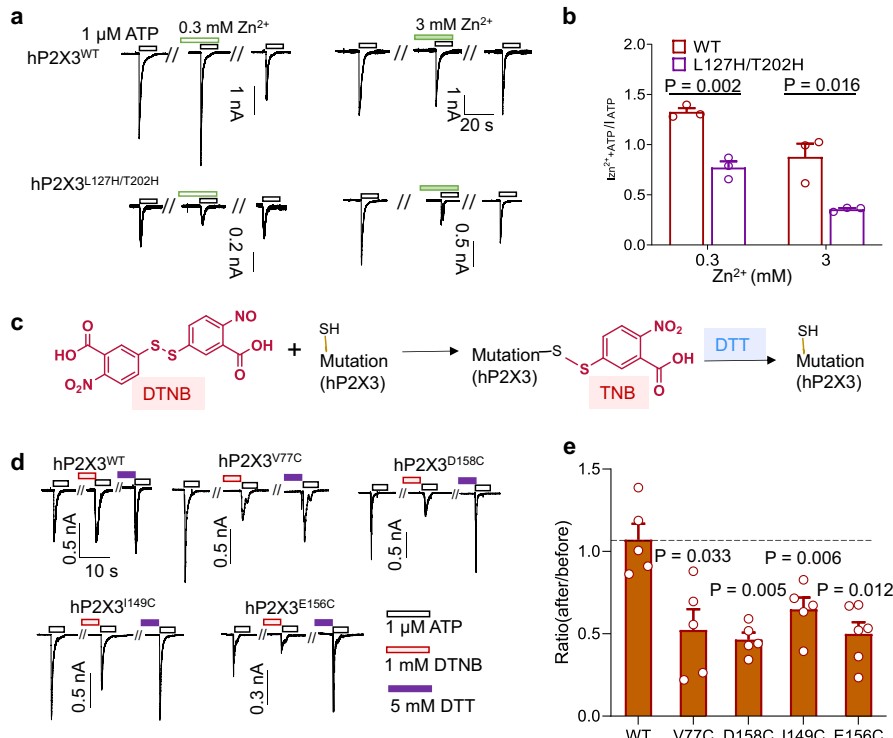

**Fig. 4 | Engineered zinc bridges and DTNB covalent occupancy affect the activation of P2X3 receptors. a, b** Representative currents (**a**) and pooled data (**b**) recorded from cells transfected with WT hP2X3 receptors and its L127H/T202H mutant. Currents were evoked by 1 μM ATP at 7 min intervals. $Zn^{2+}$ was applied at 0.3 mM (left) or 3 mM (right) as indicated. Data in **b** represent the ratio of ATP-evoked currents after to before the $Zn^{2+}$ treatment. Each circle represents an independent cell, $n = 3$. *P* value was calculated from an unpaired, two-tailed *t* test. (**c**) Chemical structure of DTNB and its modification of cysteine at the mutation site.

**d, e** Representative currents (**d**) and pooled data (**e**) recorded from cells transfected with WT P2X3 ($n = 5$) and its V77C ($n = 5$), D158C ($n = 5$), I149C ($n = 5$), and E156C ($n = 6$) mutants. Currents were evoked by 1 μM ATP at 7 min intervals. DTNB (1 mM) and DTT (5 mM) were applied as indicated. Data in **e** represent the ratio of ATP-evoked currents after to before the DTNB treatment. Each circle represents an independent cell. One sample *t* test against value 1, paired two-tailed *t* test. All summary data are expressed as mean ± SEM. Source data are provided as a Source Data file.

Since hP2X3 IP-HD contains a magnesium ion, which is coordinated by E156, D158, E111, and E109 (Fig. 1 and Supplementary Fig. 1c; green font represents the corresponding residues of the receptor that interacts with the $Mg^{2+}$), its presence will influence quercetin binding. Indeed, in the presence of 0, 1, 2, and 10 mM $Mg^{2+}$ in the standard extracellular solution, the inhibition of ATP-evoked hP2X3 currents by 10 μM quercetin reached $0.636 \pm 6.90\%$, $33.2 \pm 2.1\%$, $71.9 \pm 5.4\%$ and $75.3 \pm 5.5\%$, respectively (Supplementary Fig. 4), suggesting that $Mg^{2+}$ is required for the quercetin inhibition. Furthermore, using the extracellular domain (ECD) of hP2X3 (hP2X3$^{ECD}$) purified from *E. coli* (strain: Rosseta Origami 2) (Fig. 5c and Supplementary Fig. 5 and see Method), we determined the dissociation constants (Kd) of quercetin on P2X3$^{ECD}$ by Microscale Thermophoresis (MST), which yielded $16.0 \pm 8.5$ and $63.4 \pm 5.0$ μM in the presence and absence of $Mg^{2+}$, respectively (Fig. 5d), again demonstrating that $Mg^{2+}$ facilitates quercetin binding to hP2X3.

In order to improve binding of quercetin to hP2X3, we added sulfonic acid groups to quercetin reasoning that this could bind more tightly to $Mg^{2+}$ or adjacent positive charged residues. The newly designed synthetic compound, PSFL2915 (Fig. 5a and Supplementary Fig. 6), revealed about 15-20-fold higher apparent affinity to hP2X3 than quercetin ($IC_{50} = 0.319 \pm 0.029$ μM, Fig. 5e). It is worth noting that the addition of the sulfonic acid groups affected the apparent affinity for hP2X2 only slightly ($IC_{50} = 13.3 \pm 2.3$ μM, Fig. 5e), increasing the selectivity of PSFL2915 between P2X3 and P2X2 to about 42-fold. PSFL2915 also maintained low activities at other P2X subtypes (% inhibition at 1 μM = $49.7 \pm 7.3$, $11.8 \pm 9.2$ and $6.58 \pm 0.70$ for hP2X1, hP2X4 and hP2X7, respectively, Fig. 5e). However, it remained equipotent on the heterotrimeric rP2X2/3 ($IC_{50} = 0.261 \pm 0.042$ μM, Fig. 5e) as the homotrimeric hP2X3.

## Quercetin modulates the activation of P2X3 via IP-HD

Although the MST assay clearly demonstrated that quercetin binds to P2X3$^{ECD}$, it did not reveal if and how quercetin inhibits hP2X3 by interacting directly with IP-HD. We reasoned that if quercetin directly interacts with IP-HD, transferring the quercetin-sensitive IP-HD of hP2X3 to another P2X subtype, such as human P2X4 (hP2X4), would allow the quercetin-insensitive P2X to gain quercetin sensitivity (Fig. 6a), since IP-HD lies interior in the head domain of each subunit of the trimeric P2X receptors rather than at the subunit interface. As expected, by swapping C116-C165 of hP2X4 with the amino acids that make up the hP2X3 IP-HD (C107-C153, 47 amino acids), the chimera hP2X4$^{P2X3(C107-C153)}$ acquired the inhibition by quercetin (Fig. 6a–d). That quercetin inhibited the chimera hP2X4$^{P2X3(C107-C153)}$ ($IC_{50} = 3.61 \pm 1.11$ μM; Fig. 6d) and WT hP2X3 ($4.72 \pm 0.35$ μM; Fig. 6d) with practically identical apparent affinities suggests the IP-HD as the primary and direct site by which quercetin inhibits ATP activation of hP2X3.

To further explore how quercetin interacts with hP2X3 IP-HD, we first took the hP2X3/quercetin in silico docking pose as the initial conformation to perform conventional molecular dynamics (CMD) simulations. We found that the hydrogen bonds (H-bonds), ionic interactions, and water bridges between hP2X3, quercetin, and $Mg^{2+}$ remained stable throughout the 1.5-μs CMD simulations (Supplementary Fig. 7a–d). Then, we utilized the distance of Y114-O1...C8-OH-quercetin (CV1) and the χC2'-C1'-C2-O1 dihedral angle (Supplementary Fig. 7c) of quercetin (CV2) as two collective variables (CV) for meta-dynamics (MetaD) analysis to accelerate the conformational sampling of the hP2X3/quercetin/$Mg^{2+}$ complex (Supplementary Fig. 7c, e). The following interactions were found to be the best paradigm for the interactions between quercetin and hP2X3 IP-HD (Supplementary

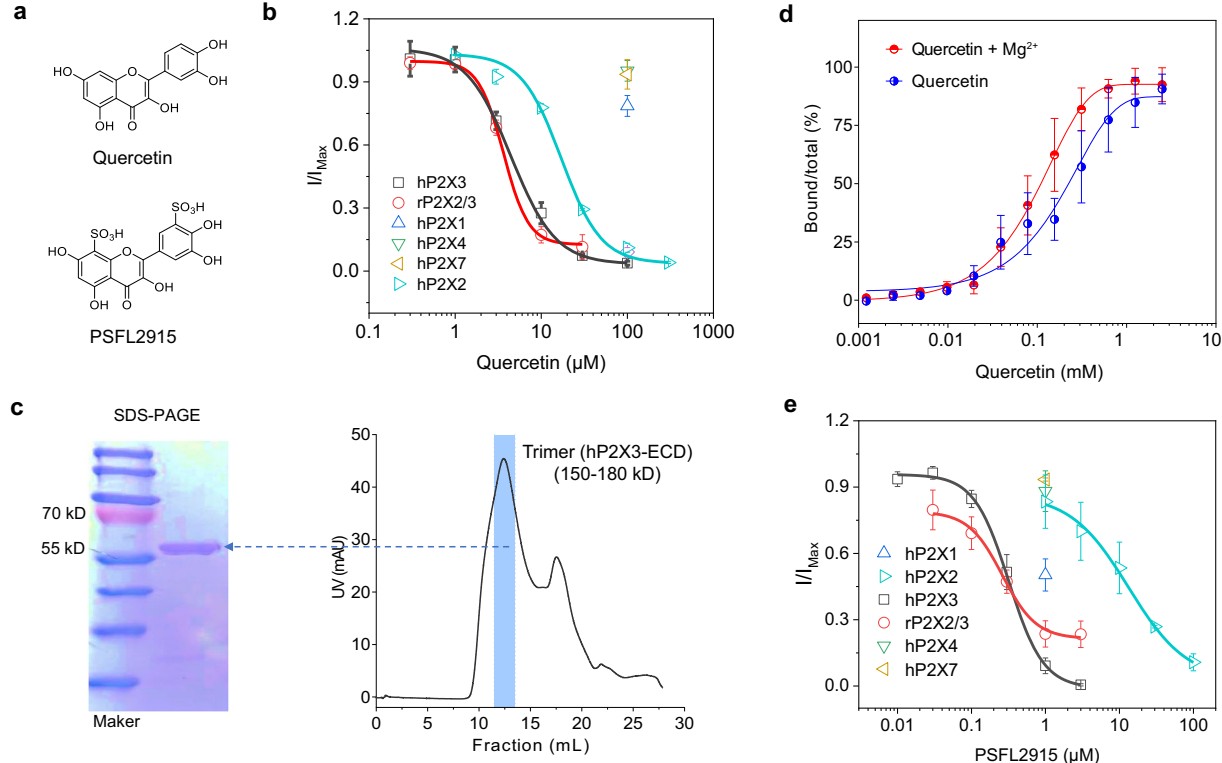

**Fig. 5 | Quercetin and its derivative PSFL2915 selectively inhibit P2X3.**
**a** Chemical structures of quercetin and PSFL2915. **b** Concentration-response curves of homomeric P2X2 and P2X3, and heteromeric P2X2/3 receptors to quercetin. Currents were evoked by 10 μM ATP for homomeric P2X2, and 1 μM ATP for homomeric P2X3 and heteromeric P2X2/3. Data were normalized to that in the absence of quercetin. Each solid line is a fit of the Hill 1 equation. $n = 3$ (0.3, 1, 30, and 100 μM) or 4 (3 and 10 μM) independent cells for hP2X3; $n = 3$ (1, 10, 30 and 300 μM), 4 (3 μM) or 7 (100 μM) independent cells for hP2X2; $n = 3$ (0.3, 1, and 10 μM) or 4 (3 and 30 μM) independent cells for rP2X2/3. In addition, ATP-evoked currents of P2X1 ($n = 6$), P2X4 ($n = 6$), and P2X7 ($n = 3$ independent cells) were tested in the presence of 100 μM quercetin. **c** Size-exclusion chromatography of the trimer of the extracellular domain of hP2X3 (hP2X3-ECD) using a GL column

(right), which was then collected for SDS-PAGE (left). The experiment was repeated thrice with similar results. **d** Binding of quercetin to hP2X3-ECD was measured by microscale thermophoresis (MST) in standard treated capillaries with and without $Mg^{2+}$ ($n = 3$). **e** Concentration-response curves of P2X2 and P2X3 homomeric receptors and P2X2/3 heteromeric receptors to PSFL2915. $n = 3$ (0.01, 0.1, and 3 μM), 4 (0.3 and 1 μM) or 5 (0.03 μM) independent cells for hP2X3; $n = 3$ (1, 30 and 100 μM) or 4 (3 and 10 μM) independent cells for hP2X2; $n = 3$ (1 μM), 4 (0.03 and 3 μM), 5 (0.3 μM) or 6 (0.1 μM) independent cells for rP2X2/3. Similar to **b**, but PSFL2915 was used. hP2X1 ($n = 4$), hP2X4 ($n = 3$) and hP2X7 ($n = 3$ independent cells) were tested with 1 μM PSFL2915. Each solid line is a fit of the Hill 1 equation. All summary data are expressed as mean ± SEM. Source data are provided as a Source Data file.

Fig. 7d, e): with the aid of $Mg^{2+}$, quercetin establishes contacts with D158, E156, and E111, and generates H-bond or hydrophobic contacts with S67, E109, K113, Y114, I149, and L297; moreover, $Mg^{2+}$ also maintains the conformation of IP-HD by coming into direct contacts with D158, E156, E111, and E109 (Supplementary Fig. 1c).

To validate the above prediction, single point mutations were made on hP2X3 and tested for quercetin inhibition of ATP-evoked currents. The inhibitory action of 10 μM quercetin was substantially attenuated in hP2X3$^{S67F \ or \ S67W}$, hP2X3$^{E109A}$, hP2X3$^{S110F}$, hP2X3$^{E111I}$, hP2X3$^{K113F}$, hP2X3$^{Y114F}$, hP2X3$^{G129P}$, hP2X3$^{I149F}$, hP2X3$^{E156A \ or \ E156F}$, hP2X3$^{D158A}$, and hP2X3$^{L297A \ or \ L297W}$ (Fig. 6e, f), supporting the prediction by CMD and MetaD. In particular, the IC$_{50}$ of quercetin increased ~10-50-fold for hP2X3$^{L297A}$ (210 ± 4 μM), hP2X3$^{E109A}$ (197 ± 166 μM), hP2X3$^{S67F}$ (68.9 ± 12.7 μM), hP2X3$^{E156A}$ (48.3 ± 13.6 μM), and hP2X3$^{G129P}$ (41.7 ± 27.9 μM) when compared to WT hP2X3 (4.72 ± 0.35 μM, Fig. 6e–g). Meanwhile, we ruled out two other potential sites proximal to IP-HD, the central pocket and the pocket formed by the β2,3-sheet and head domain, since hP2X3$^{Y285A}$, hP2X3$^{K284R}$, hP2X3$^{K284A}$, hP2X3$^{F282W}$, hP2X3$^{G130A}$, hP2X3$^{L133F}$, hP2X3$^{G66A}$, hP2X3$^{L69W}$, hP2X3$^{R73A}$, hP2X3$^{S78G}$, hP2X3$^{P128V}$, hP2X3$^{A168F}$ and hP2X3$^{I132F}$ displayed similar degrees of quercetin inhibition as WT hP2X3 (Supplementary Fig. 8), although hP2X3$^{K284R}$ exhibited slightly decreased inhibition by quercetin.

Likewise, the inhibition by quercetin was considerably reduced by covalent occupancy of an engineered cysteine inside IP-HD, hP2X3$^{S67C}$,

with an irreversible alkylation reagent (NPM, 2 mM) (Fig. 7a–c). Moreover, to perturb the conformation dynamics of IP-HD, mutants with the engineered disulfide bond, hP2X3$^{D158C/E111C}$ and hP2X3$^{L127C/T202C}$, were employed (Fig. 3a). After crosslinking with 0.3% $H_2O_2$, the ability of quercetin to inhibit the ATP-evoked currents of these two mutants was significantly reduced compared to the treatment with 10 mM DTT (Fig. 7d, e). Altogether, the above results lend strong support to the idea that quercetin inhibits hP2X3 by acting at IP-HD.

## PSFL2915 prevents the allosteric change of IP-HD

Because PSFL2915 contains two additional negatively charged sulfate groups compared to quercetin, it could interact with positively charged residues like Lys and Arg within the ATP-binding pocket more than with IP-HD of hP2X3. We ruled out this possibility because of the following findings. First, the ability of PSFL2915 to inhibit hP2X3 currents activated by 1, 10, and 100 μM ATP did not change significantly (91.5 ± 2.8%, 90.2 ± 3.5%, and 88.7 ± 3.4% inhibitions, respectively), demonstrating that it is a noncompetitive inhibitor (Fig. 8a). Second, covalent occupancy of IP-HD in P2X3$^{S67C}$ significantly suppressed the inhibitory effect of PSFL2915 (Fig. 8b), suggesting that the drug needs to gain access to IP-HD. Third, at 1 μM, PSFL2915 did not inhibit hP2X4, but it produced 86.9 ± 3.7% inhibition of the hP2X4$^{P2X3 (C107-C153)}$ chimera which has the ATP binding site of P2X4 but the IP-HD of P2X3 (Fig. 8c). Fourth, the inhibitory effect of PSFL2915 is $Mg^{2+}$ dependent. With 0, 1,

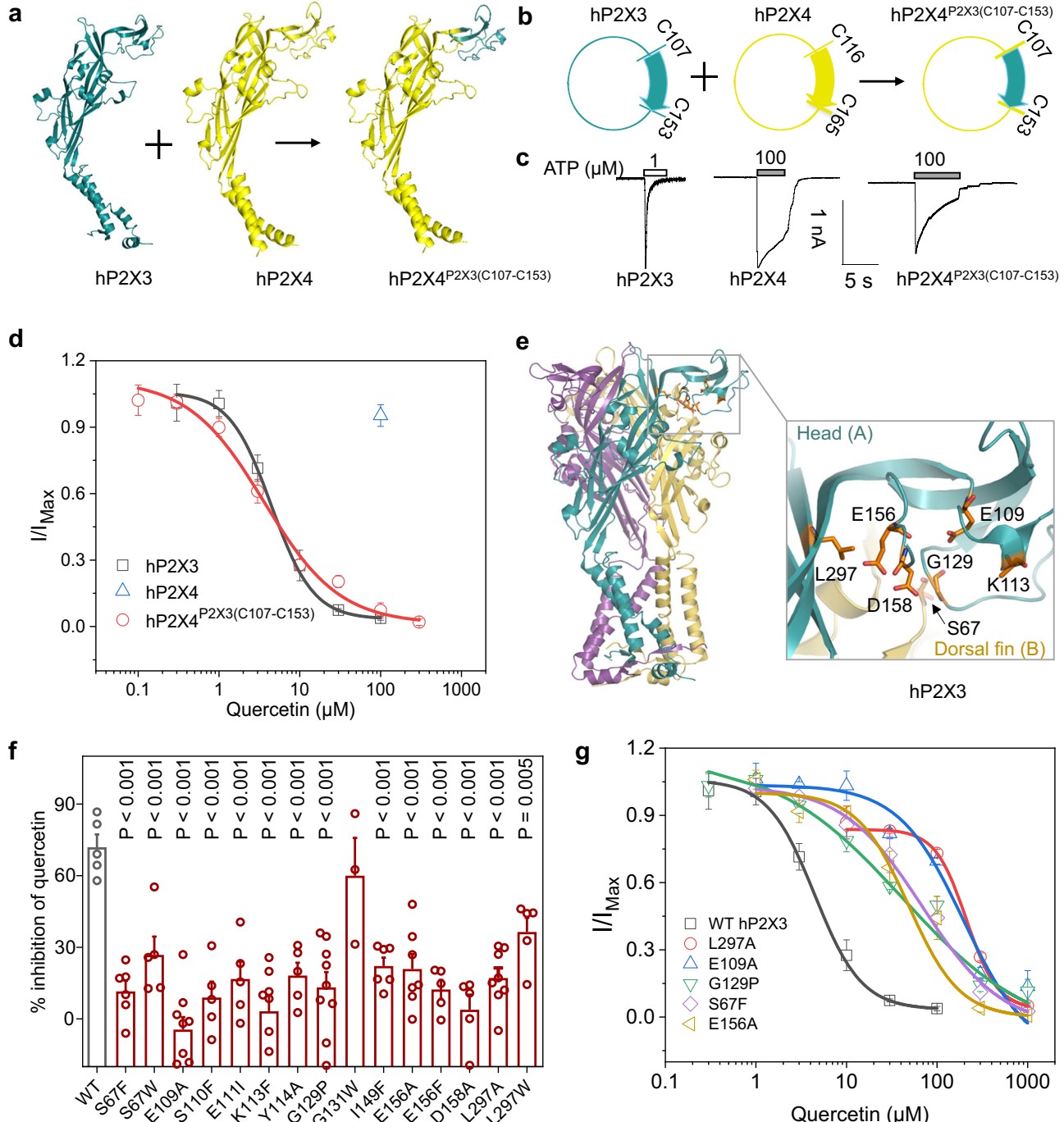

**Fig. 6 | Quercetin inhibits the activation of hP2X3 via acting at IP-HD.**
**a**, **b** Swapping the IP-HD of hP2X3 to hP2X4 allows hP2X4 to acquire inhibition by quercetin. C116-C165 of hP2X4 was replaced by the corresponding C107-C153 fragment of hP2X3 to produce the chimera hP2X4$^{P2X3 (C107-C153)}$. **c** Representative currents evoked by 1 μM ATP for hP2X3, and 100 μM ATP for hP2X4 and hP2X4$^{P2X3(C107-C153)}$. **d** Concentration-response curves to quercetin of hP2X3 and hP2X4$^{P2X3(C107-C153)}$. hP2X4 was tested at 100 μM quercetin, each solid line was a fit of the Hill 1 equation. $n = 3$ (0.3, 1, 30 and 100 μM) or 4 (3 and 10 μM) independent cells for hP2X3; $n = 6$ independent cells for hP2X4; $n = 3$ (0.1, 100, and 300 μM), 5 (0.3 μM) or 7 (1, 3, 10, and 30 μM) independent cells for hP2X4$^{P2X3 (C107-C153)}$.
**e** Zoomed-in view of the IP-HD of hP2X3. Residues around this pocket are indicated with sticks for emphasis. **f** Effect of quercetin (10 μM) on ATP (1 μM)-induced activation of hP2X3-WT and the indicated mutants. Each circle represents an independent cell; $n = 3$ for G131W; $n = 5$ for WT, S67W, S110F, E111I, Y114A, E156F,

D158A, and L297W; $n = 6$ for S67F and I149F; $n = 7$ for E156A; $n = 8$ for K113F and L297W; $n = 9$ for E109A and G129P. $P$ value was compared to WT, one-way ANOVA followed by Dunnett's multiple comparisons test, $F (15, 80) = 8.733$.
**g** Concentration-response curves to quercetin of WT hP2X3 and several of its mutants (solid lines were fits of the Hill 1 equation). $n = 3$ (0.3, 1, 30 and 100 μM) or 4 (3 and 10 μM) independent cells for WT; $n = 3$ (1, 3, 10, 100, 300, and 1000 μM) or 4 (30 μM) independent cells for S67F; $n = 3$ (3, 10, 30, and 100μM) or 4 (1, 300, and 1000 μM) independent cells for E109A; $n = 3$ (0.3, 30, and 1000 μM), 4 (1 and 300 μM), 5 (3 μM) or 6 (10 and 100 μM) independent cells for G129P; $n = 3$ (1000 μM), 4 (3, 100, and 300 μM), 5 (10 and 30 μM) or 6 (1 μM) independent cells for E156A; $n = 3$ (30, 300, and 1000 μM), 4 (10 μM) or 5 (100 μM) independent cells for L297A. All summary data are expressed as mean ± SEM. Source data are provided as a Source Data file.

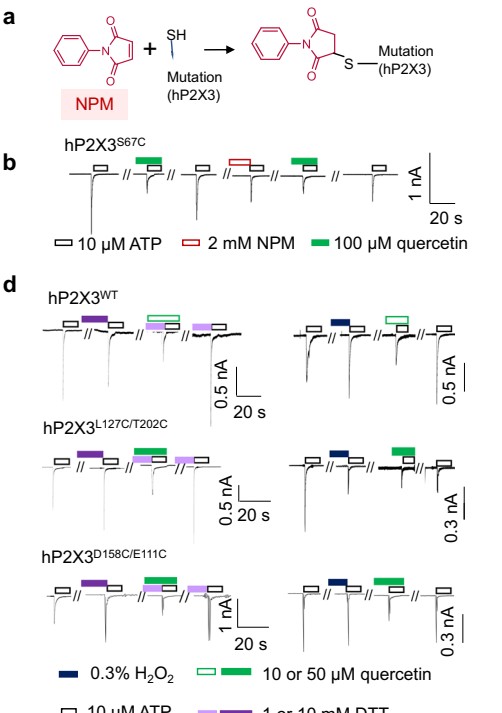

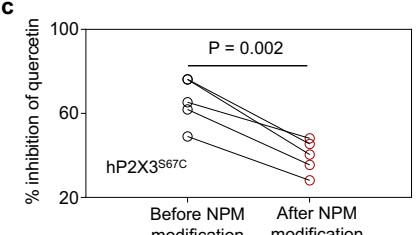

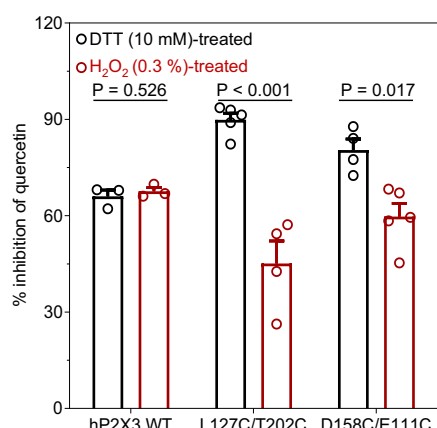

**Fig. 7 | NPM covalent occupancy and disulfide crosslinking affect the inhibitory effect of quercetin on hP2X3. a** Chemical structure of NPM and its modification of cysteine at the mutation site. **b** Representative currents recorded from cells transfected with P2X3$^{S67C}$ receptors. Cells were voltage-clamped at −60 mV and currents were evoked by ATP (10 μM) at 7 min intervals. NPM (2 mM) and quercetin (100 μM) were applied as indicated. **c** Pooled data for experiments in **b**. The inhibition rates of quercetin before and after the NPM treatments for individual cells are connected with lines. Each line represents an independent cell, $n = 5$. $P$ value was calculated from a paired, two-tailed $t$ test. **d** Representative currents recorded from cells transfected with WT hP2X3 and its double cysteine mutants, D158C/E111C and L127C/T202C. ATP, H$_2$O$_2$, DTT, and quercetin were applied as indicated. **e** Pooled data for experiments in **d**. Each circle represents an independent cell, $n = 3$ for WT hP2X3, $n = 5$ for L127C/T202C (DTT treated), $n = 4$ for L127C/T202C (H$_2$O$_2$ treated), $n = 4$ for D158C/E111C (DTT treated), $n = 5$ for D158C/E111C (H$_2$O$_2$ treated). $P$ value was calculated from an unpaired, two-tailed $t$ test. All summary data are presented as mean ± SEM. Source data are provided as a Source Data file.

2, and 10 mM Mg$^{2+}$ in SS, 0.6 μM PSFL2915 inhibited hP2X3 by 17.9 ± 4.3%, 83.5 ± 3.1%, 86.3 ± 2.2% and 89.3 ± 2.8%, respectively (Supplementary Fig. 4). These results confirm that like quercetin, PSFL2915 also acts at the IP-HD site.

Because of the autofluorescence owing to the presence of chromogenic groups and the relatively low affinity, the high concentrations of quercetin (~10–50 μM) sufficient to inhibit hP2X3 tended to interfere with the ANAP emission spectrum measurement. However, with the increased affinity, PSFL2915 could be used at a much lower concentration (0.5 μM) to significantly inhibit hP2X3 activation without causing a notable change in the emission spectrum measured using the excitation filter for ANAP in either the absence or presence of 10 μM ATP (Fig. 8d, e). Thus, we could assess how the inhibitor affects the conformational dynamics in this region by comparing ATP-induced peak emission wavelength shift of ANAP incorporated into IP-HD before and after PSFL2915 applications.

As shown earlier, a number of sites within and outside the hP2X3 IP-HD, when incorporated with flUAA ANAP, underwent significant peak emission wavelength shifts in response to ATP administration in the VCF analysis, implicating changes in their microenvironment during channel activation (Fig. 2c). For the sites inside the IP-HD (e.g., Y114, E111, G131, E156, and so on), the ANAP substitution most likely would interfere with PSFL2915 binding. Therefore, we considered the two mutants with ANAP incorporated outside the IP-HD to examine the effect of PSFL2915 on the ATP-induced conformation dynamics of IP-HD. The substitution at R281 was excluded as it is directly involved in ATP-binding. Thus, we used hP2X3$^{L297-ANAP-YFP}$ for the VCF analysis. L297 is situated on a rigid β-strand and juxtaposed with the region of IP-HD; the emission spectrum of ANAP inserted at this position is passively

shifted due to the significant leftward shift and squeezing of loops 108–118 and 122–134 in lower head domain after ATP binding (Figs. 2c, 6e, 8f, and Supplementary Fig. 1b, c). Additionally, unlike hP2X3$^{S67C}$, covalent modification of hP2X3$^{L297C}$ by DTNB had no discernible impact on the inhibition caused by quercetin (Fig. 8g), which indicates that this site can tolerate insertion of a bulkier side chain (this is also evidenced by the fact that L297W only slightly affects the inhibition efficiency of quercetin; Fig. 6f). Indeed, the treatment of PSFL2915 nearly completely abolished the ATP-induced peak emission wavelength shift of hP2X3$^{L297-ANAP-YFP}$ (Fig. 8h–j), consistent with the idea that the inhibitor binding at the IP-HD region blocks the conformational changes induced by ATP.

## Quercetin alleviates chronic cough via P2X3

P2X3 is a target for the treatment of chronic cough[22,23,54], and quercetin can alleviate cough symptoms[17,21,53]. However, it is unclear whether quercetin suppresses cough by targeting P2X3 on top of its anti-inflammatory properties. To answer this question, we first employed a chronic cough model in C57BL/6 mice induced by 0.1% ammonia aerosol (see Methods and Fig. 9a). Based on the most common oral doses of quercetin used in humans, i.e., between 0.5 and 2.4 g per day[55], we estimated the equivalent dose in mice using a body surface area discounting algorithm[56], taking into account the volume and solubility of the drug in mice, to be 50 mg/kg and 150 mg/kg by oral gavage twice daily (b.i.d.). We found that 150 mg/kg quercetin (b.i.d.) significantly decreased the frequency of coughing in mice (Fig. 9b). However, 50 mg/kg quercetin (b.i.d.) did not (Fig. 9b). In addition, PSFL2915 significantly decreased mouse coughing when applied at a dosage of 15 mg/kg (b.i.d.), which was

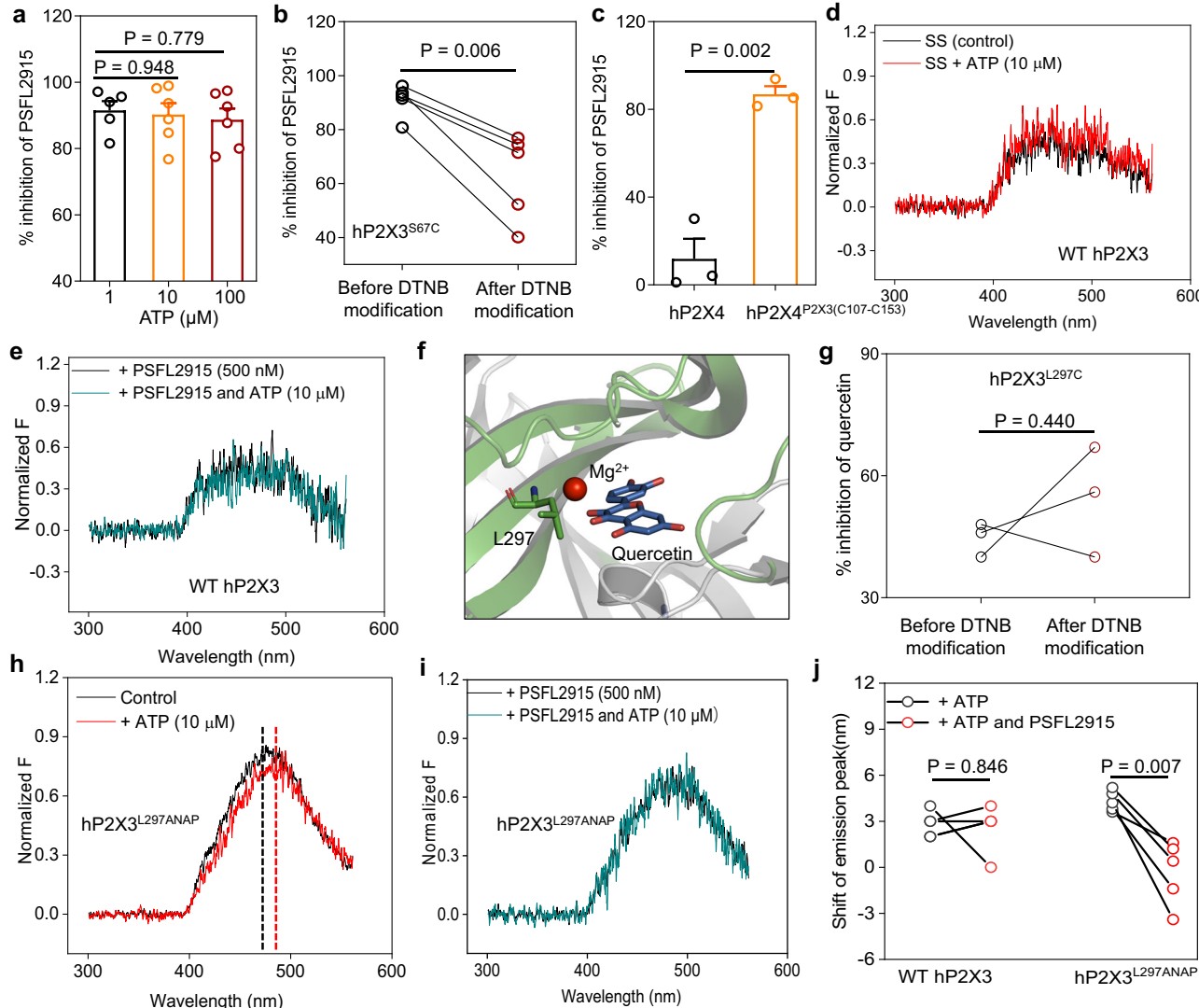

**Fig. 8 | PSFL2915 blocks the conformational changes of IP-HD in P2X3 induced by ATP binding. a** Effect of PSFL2915 (1 μM) on ATP (1, 10, 100 μM)-induced activation of WT hP2X3. Each circle represents an independent cell. $n = 5$ (1 μM) or 6 (10 and 100 μM); one-way ANOVA and Dunnett's multiple comparison test, F (2, 14) = 0.8142. Data are expressed as mean ± SEM. **b** DTNB (2 mM) modification reduced the inhibition of hP2X3$^{S67C}$ by PSFL2915 (1 μM). Each line represents an independent cell, $n = 5$. $P$ value was calculated from a paired, two-tailed $t$ test. **c** Effect of PSFL2915 (1 μM) on ATP (100 μM) induced activation of hP2X4 and hP2X4$^{P2X3 (CI07-CI53)}$. Each circle represents an independent cell; $n = 3$. $P$ value was calculated from an unpaired, two-tailed $t$ test. Data are expressed as mean ± SEM. **d, e** Representative emission spectra in the absence and presence of ATP (10 μM)

for WT hP2X3. **f** Zoomed-in view of the location of L297 and bound quercetin. **g** DTNB (2 mM) modification did not reduce the inhibition rate of hP2X3$^{L297C}$ by quercetin (10 μM). Each line represents an independent cell, $n = 3$; paired two-tailed $t$ test. **h, i** Representative emission spectra in the absence and presence of ATP (10 μM) for the ATP-induced shift of the peak emission wavelengths for hP2X3$^{L297ANAP}$. **d, e, h, i** PSFL2915 (500 nM) was applied as indicated. Black, control; red, emission spectrum in the presence of 10 μM ATP; dark cyan, emission spectra in the presence of 10 μM ATP and 500 nM PSFL2915. **j** Pooled data for experiments in **d, e, h, i**. Each line represents an independent cell; $n = 5$. $P$ value was calculated from a paired, two-tailed $t$ test. Source data are provided as a Source Data file.

comparable to an equivalent dose of gefapixant (15 mg/kg, b.i.d., Fig. 9c).

Consistent with that P2X3 plays an important role in coughing, *P2rX3*$^{-/-}$ mice coughed much less frequently than *P2rX3*$^{+/+}$ mice (Fig. 9d). Oral dosing of quercetin (150 mg/kg, b.i.d.) reduced the frequency of coughing in *P2rX3*$^{+/+}$ mice (Fig. 9e), but did not further reduce that in *P2rX3*$^{-/-}$ mice (Fig. 9e), suggesting that quercetin alleviates coughing by acting at P2X3.

Next, using the conventional Hartley guinea pig cough model induced by citric acid aerosol and ATP, we further investigated the cough-suppressant effects of quercetin, PSFL2915, and gefapixant (Fig. 9f). We found that, when given twice daily for seven days, quercetin (100 mg/kg) or PSFL2915 (10 mg/kg) reduced the frequency of coughing in guinea pigs as well as gefapixant (10 mg/kg) (Fig. 9f);

however, 25 mg/kg of quercetin, b.i.d., had no significant effect (Fig. 9f).

Thus, both quercetin and PSFL2915 reduced cough frequency in mice and the conventional Hartley guinea pig cough models. Furthermore, at equivalent dosages, PSFL2915 and the known P2X3-targeting cough-suppressant gefapixant had roughly the same effect. For quercetin, higher doses than that of PSFL2915 and gefapixant are needed.

**Quercetin and PSFL2915 do not cause taste disturbances**

Gefapixant is approved as a cough-suppressant despite having a significant risk of causing taste anomalies[12]. To evaluate if quercetin and PSFL2915 also have the similar side effect as gefapixant, we tested the impact of quercetin and PSFL2915 on the ability of mice to detect

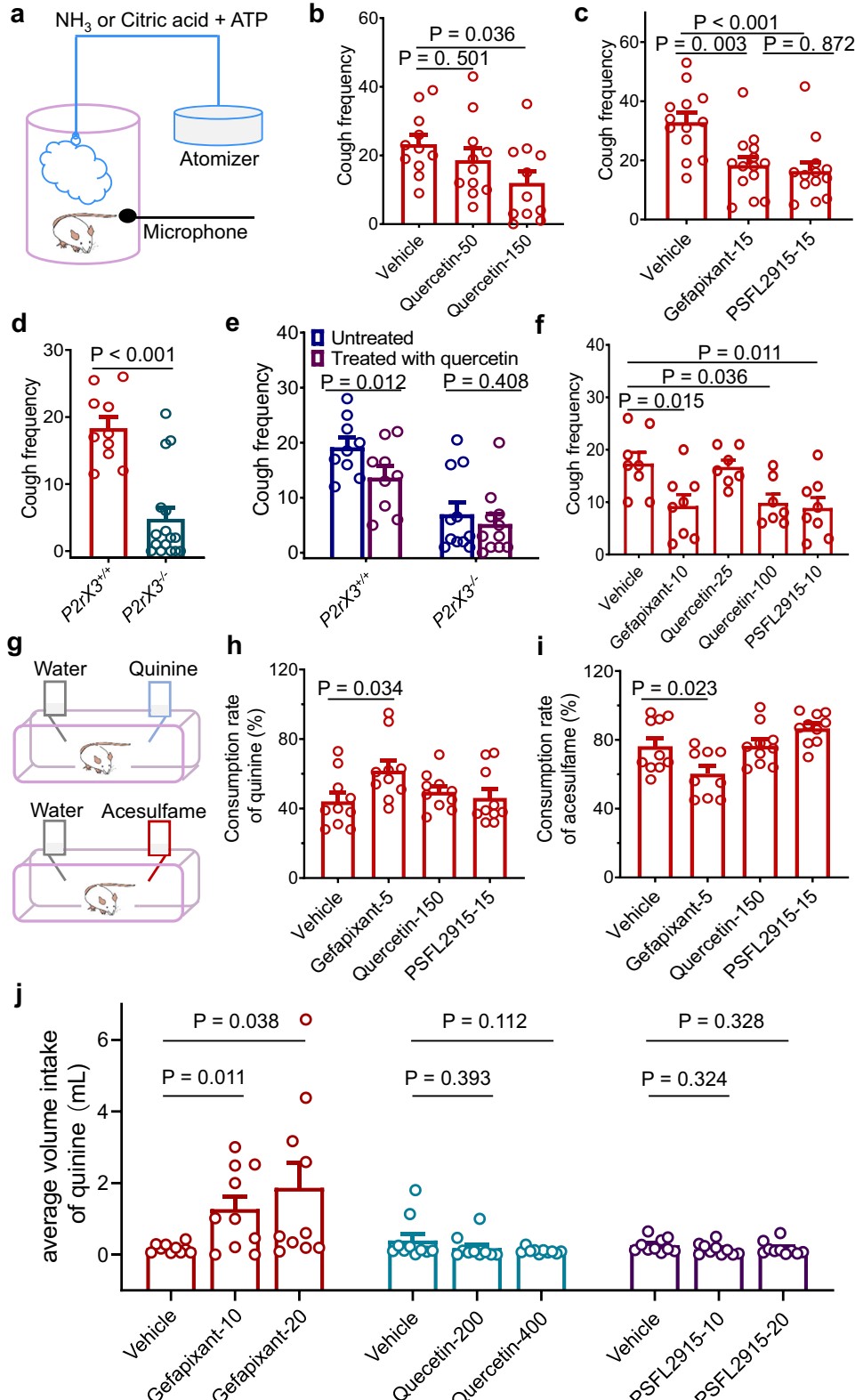

bitterness (0.3 mM quinine)[57] using a two-bottle drink paradigm (Fig. 9g and see Methods). In line with its taste disturbing effect, oral dosing of gefapixant at 1/3 of the dose used for cough suppression, i.e., 5 mg/kg (b.i.d.), resulted in a significantly higher proportion of quinine consumption than in the vehicle control group (61.8 ± 5.6% vs. 44.1 ± 4.9%, gefapixant vs. control (vehicle = 0.5% CMC-Na); Fig. 9h). However, neither quercetin (150 mg/kg, b.i.d.) nor PSFL2915 (15 mg/kg,

b.i.d.) caused any discernible taste disturbance in mice (49.6 ± 3.3% for quercetin, and 46.1 ± 5.0% for PSFL2915 vs. 44.1 ± 4.9% for blank/ water; Fig. 9h).

Because bitterness is regarded to be a kind of punishment, we repeated the experiment with a sweetener (acesulfame-K, 0.5 mM)[58]. As expected, mice administered with gefapixant (5 mg/kg, b.i.d.) drank acesulfame-K solution at a significantly lower rate than those in the

**Fig. 9 | Quercetin and PSFL2915 reduce cough in C57BL/6 mice and guinea pigs without taste impairments. a** Schematic diagram of the experimental setup. **b** Cough frequency in mice administered with vehicle (0.5% CMC-Na) or quercetin (50 or 150 mg/kg, b.i.d.). Coughing was induced by $NH_3$. $n = 11$, One-way ANOVA and Dunnett's multiple comparison tests, $F_{(2, 30)} = 3.070$. **c** Cough frequency in mice administered vehicle ($n = 13$), gefapixant (15 mg/kg, b.i.d., $n = 14$), or PSFL2915 (15 mg/kg, b.i.d., $n = 13$). Coughing was induced by $NH_3$ (0.1%). One-way ANOVA followed by Tukey's multiple comparisons test, $F_{(2, 37)} = 9.467$. **d** Cough frequency of $P2rX3^{+/+}$ ($n = 10$) and $P2rX3^{-/-}$ ($n = 16$) mice that received $NH_3$, unpaired two-tailed $t$ test. **e** Cough frequency $P2rX3^{+/+}$ ($n = 9$) and $P2rX3^{-/-}$ ($n = 11$) mice untreated or treated with quercetin (150 mg/kg, b.i.d.) before the exposure to $NH_3$. Paired two-tailed $t$ test. **f** Cough frequency in guinea pigs treated with vehicle ($n = 8$), gefapixant (10 mg/kg, b.i.d., $n = 8$), PSFL2915 (10 mg/kg, b.i.d., $n = 8$), or quercetin (25 or 100 mg/kg, b.i.d., $n = 7$). Coughing was induced by citric acid and ATP. One-way

ANOVA followed by Dunnett's multiple comparisons test, $F_{(4, 33)} = 4.918$. **g** Schematic diagram of the two-bottle preference test setup. **h, i** Two-bottle preference tests of mice. Compared to vehicle control mice ($n = 10$), gefapixant (5 mg/kg, b.i.d.) treated mice drank more bitter quinine ($n = 10$) and less sweetener acesulfame ($n = 9$), whereas PSFL2915 (15 mg/kg, b.i.d., $n = 10$) or quercetin (150 mg/kg, b.i.d., $n = 10$) treatment resulted neither increased quinine intake nor decreased acesulfame intake. One-way ANOVA followed by Dunnett's multiple comparisons test, $F_{(3, 36)} = 2.762$ (**h**) and $F_{(3, 35)} = 7.141$ (**i**). **j** Two-bottle preference tests in rats. Rats treated with gefapixant (10 and 20 mg/kg) drank more quinine solution compared to vehicle controls, whereas PSFL2915 (10 and 20 mg/kg) or quercetin (200 and 400 mg/kg) treatments did not result in increased quinine intake. Paired two-tailed $t$ test; $n = 10$. All summary data are expressed as mean ± SEM, and each circle represents an independent animal. Source data are provided as a Source Data file.

control group ($60.3 \pm 4.6\%$ vs. $76.3 \pm 4.8\%$, gefapixant vs. control; Fig. 9i). However, the administration of quercetin (150 mg/kg, b.i.d.) or PSFL2915 (15 mg/kg, b.i.d.) did not alter mice in taking the acesulfame-K solution ($76.5 \pm 3.7\%$ for quercetin, and $86.7 \pm 2.8\%$ for PSFL2915 vs. $76.3 \pm 4.8\%$ for control; Fig. 9i).

We also tested the impact of quercetin and PSFL2915 on the ability of rats to perceive bitterness (0.3 mM quinine) using a two-bottle drink paradigm (see Methods). The volume of quinine solution (bitter sensation) consumed by the rats increased significantly after intraperitoneal injection of 10 and 20 mg/kg of gefapixant (Fig. 9j). In contrast, the volume of quinine solution consumed by the rats did not change significantly after intraperitoneal injection of 200 and 400 mg/kg of quercetin (Fig. 9j) or 10-20 mg/kg PSFL2915 (Fig. 9j). Together, these results suggest that quercetin and PSFL2915 have no taste impairment in rodents.

In light of the aforementioned results, at least for PSFL2915, an allosteric inhibitor acting at hP2X3 IP-HD has the same cough-suppressing potency as gefapixant but does not affect taste perception in mice. This has tremendously intriguing implications for the development of innovative cough-suppressant drugs that target P2X3 (Fig. 10).

## Discussion

We show that the activation of P2X3 differs from that of P2X2 by necessitating a leftward shift of loops of the lower head domain, causing a tightening of the IP-HD pocket (Fig. 10)[48]. Not only are the conformational changes in this region during P2X3 opening directly observed by flUAA incorporation and VCF analysis, but P2X3 activation is inhibited by preventing IP-HD tightening using multiple approaches, including designed disulfide crosslinking, covalent occupancy, and engineered zinc ion bridges. This unique allosteric regulatory site may be exploited for the treatment of chronic cough, as small molecule bound to this pocket not only suppressed P2X3 activation by inhibiting IP-HD tightening but also alleviated cough in both mice and guinea pigs. To the best of our knowledge, this is the first claim that compounds could modulate the allosteric change of IP-HD in P2X receptors, which differs from another druggable allosteric site, fostered by LF, LB, and DF domains, bound by gefapixant[12,59]. In addition to demonstrating P2X3 as an in vivo target of quercetin, we also made a synthetic compound, PSFL2915, which exhibits equipotent cough-suppressing effect as gefapixant. Notably, unlike gefapixant, targeting the IP-HD as an allosteric site (at least for quercetin and PSFL2915) produces no adverse effect on mammalian taste perception, making it especially suitable for the development of P2X3-based cough suppressants (Fig. 10 and ref. 54). Our findings in mice are consistent with the lack of taste-related adverse effects in clinical trials utilizing quercetin at a dose of 5 g/day for 28 days[60], as well as the lack of adverse effects associated with taking quercetin as a dietary supplement[55,61].

The ATP recognition site of the P2X receptor is positioned in a pocket surrounded by the head, LF, DF, and lower body domains. ATP forms H-bonds or ionic interactions with conserved amino acid residues K67, K69, T189, N293, R295, and K313 (numbered after hP2X4) within this pocket. Other nearby amino acid residues outside this pocket are not conserved[42], for example, the amino acids in the head domain are nonconserved (Supplementary Fig. 2a, b). The relative motion of the LF and DF domains is critical for P2X function[62,63]. The LF region is conserved at both ends but not in the middle of the sequence (Supplementary Fig. 2a, b), and this difference in the number and type of amino acids in the middle distinguishes the LF domain's interaction with other regions in different subtypes. S275, for example, is involved in the formation of the ATP-binding pocket in P2X3, but not other P2X subtypes, and the S275A mutation could speed up the recovery time after channel desensitization only in P2X3[64]. Due to sequence variations in the head, DF, and LF domains among different P2X subtypes, as well as the intricate gating mechanism involving interactions between neighboring regions, opportunities exist for developing subtype-specific small molecule probes based on pockets formed by these regions, such as IP-HD.

Refractory chronic cough (RCC) is a kind of cough that has no clear etiological factors, lasts more than eight weeks, is not treated for long-term, and has a complex pathophysiology[65]. Because RCC may be caused by airway inflammation, glucocorticoids such as mometasone, budesonide, and beclomethasone are also used to treat it[66]. Cough reflex hypersensitivity, which comprises increased peripheral and central sensitization of the cough reflex[66], is one of the key hallmarks of RCC. Both dextromethorphan, a central cough-suppressant that inhibits the medullary cough center, and narcodine, a peripheral cough-suppressant that blocks cough sensors, afferent or efferent neurons, and so on, may reduce cough frequency in RCC, but their usage is limited due to their side effects[67]. Sensory neuron hypersensitization produced by P2X3 overexpression in airway vagal afferent neurons results in pathogenesis of chronic cough[59]. The P2X3 receptor is presently the sole new target established via proof-of-concept (POC) for the treatment of RCC, and the success of the clinical trial on gefapixant supports this idea[59]. Consistently, $P2rX3^{-/-}$ mice developed significantly less number of coughs than their WT littermates in the rodent cough model (Fig. 9d).

Quercetin is a common flavonoid with a polyphenolic hydroxyl structure found in many foods, such as onions, fine onions, asparagus, cabbage, mustard, green peppers, chicory, grapefruit, lettuce, hawthorn, apples, mangoes, plums, radishes, blackcurrants, potatoes, and spinach (Fig. 10). It offers natural antioxidant activity and is widely promoted as a dietary supplement, with health benefits such as improvement of pulmonary nodules and cough symptoms alleviation[17,68]. Some herbal cough-suppressant drugs contain quercetin[69], and clinical trials have shown that quercetin can ameliorate cough symptoms in COVID-19 patients[20,21]. However, it is not clear

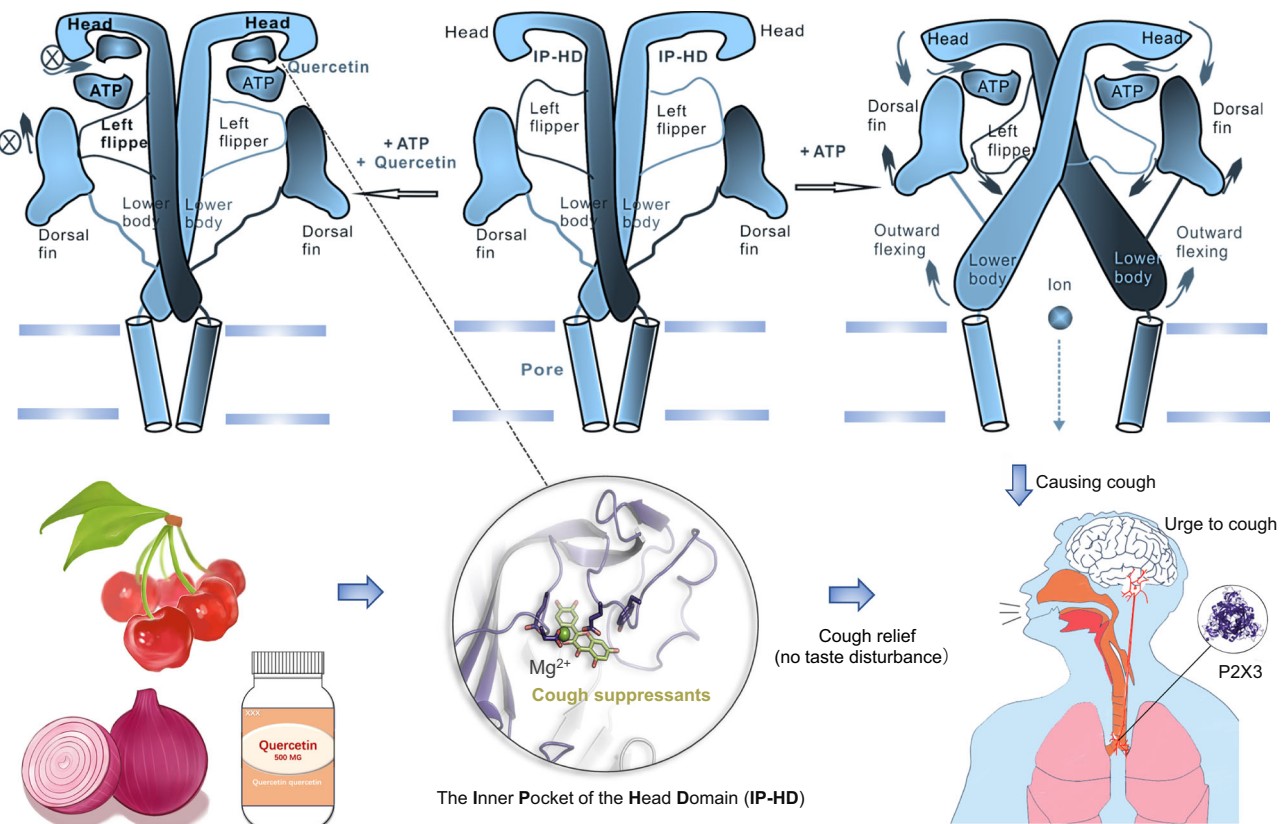

**Fig. 10 | Conformational changes of IP-HD during P2X3 receptor channel gating, and schematic diagram of the mechanism of inhibition by quercetin and PSFL2915.** Tightening of IP-HD is essential for P2X3 receptor activation, and quercetin from natural products inhibits P2X3 receptor activation by blocking the tightening of IP-HD. This likely underlies the main mechanism of quercetin and PSFL2915 in cough relief.

whether quercetin exerts this antitussive effect directly or indirectly by ameliorating other diseases. We show here that quercetin is efficient at relieving cough in mice and guinea pigs (Fig. 9b, f). Although quercetin has anti-inflammatory properties, there is little evidence that this is the principal mechanism by which it suppresses cough symptoms[18,70]. That quercetin reduces the cough frequency of *P2rX3*[+/+], but not *P2rX3*[−/−], mice (Fig. 9e) suggests that quercetin suppresses cough via acting at P2X3. Furthermore, by increasing the affinity for P2X3 through the introduction of sulfonic acid groups but decreasing the antioxidant capacity of phenolic hydroxyl groups, the cough-suppressing effect of PSFL2915 is significantly increased compared to quercetin (Fig. 9f), again supporting that quercetin exerts its anti-cough action (at least mainly) by blocking P2X3 receptors, in addition to non-specific mechanisms such as antioxidant effects.

The flavonoid quercetin is often shown to be one of the active molecules in high-throughput screening, but some factors can lead to false positives during screening, such as its fluorescence, membrane perturbation, antioxidant activity and metabolites[71]. Among them, fluorescence and membrane perturbation are mainly associated with false positives in in vitro activity screening, while antioxidant activity and its metabolism are mainly associated with false positives in in vivo screening. Here, the inhibitory activity of quercetin on the P2X3 receptor was determined electrophysiologically which allows for direct measurement of the magnitude of the current, independent of fluorescence; mutation of the P2X3 receptor directly eliminated the inhibitory effect of quercetin on P2X3, and the chimeric construction of the IP-HD endowed the inhibitory effect of quercetin to be accessed through the P2X4, which also demonstrates that quercetin is acting directly on the P2X3 receptor. The lack of efficacy of quercetin in *P2rX3*[−/−] mice essentially establishes that the cough-suppressant effect of quercetin is directly related to the P2X3 receptor, although the

contribution to the cough-suppressant effect from antioxidant activity and its metabolites through other pathways, cannot be completely excluded.

Finally, despite the therapeutic advantage that gefapixant may provide for RCC patients, its clinical applicability will be limited due to the high prevalence of taste disorders[59]. This is thought to be due to gefapixant's poor selectivity (-3-7 times) for P2X3 (homotrimer) and P2X2/3 (heterotrimer). P2X3 is mostly found as a homotrimer in the respiratory system, but more often as a P2X2/3 heterotrimer in the gustatory system[72]. It is suggested that gefapixant suppresses cough by inhibiting P2X3 on respiratory neurons while simultaneously suppressing P2X2/3 on gustatory receptors for its side effect on taste abnormalities[23]. However, neither quercetin nor PSFL2915 is selective between P2X3 and P2X2/3 (Fig. 5b, e), but no significant taste dysfunction developed in animals treated with these drugs (Fig. 9h–j). Additionally, human and animal safety studies also confirmed that long-term, higher doses of quercetin do not produce taste disorders[55]. The lack of side effects of quercetin and PSFL2915 on taste might be explained by the following. First, this may be related to quercetin's $Mg^{2+}$ dependence (Supplementary Fig. 4b). One of the main causes of coughing is the release of intracellular ATP to the extracellular space and activation of P2X3 in respiratory neurons in response to external stimuli[12,23]. Coincidentally, because intracellular $Mg^{2+}$ concentration is about 20 times of that in tissue fluid[73,74], it is likely that intracellular $Mg^{2+}$ is also released to the extracellular space together with ATP, increasing the affinity of quercetin for P2X3. Secondly, it may be related to the in vivo distribution of quercetin after administration. Experiments in rodents have shown that quercetin is 4–8 times more preferentially distributed to the lung than muscles (the tongue, where the taste buds are located, is primarily a muscle)[75,76], which would favor the inhibition of P2X3 more in the respiratory system than that of

P2X2/3 in the tongue by quercetin. More studies are needed to clarify why quercetin and PSFL2915 do not disturb taste function.

In conclusion, we provide evidence that IP-HD tightening is crucial for P2X3 channel opening, and based on this mechanism, inhibitors targeting P2X3-specific allosteric modulation at IP-HD were found to have cough-suppressing effects without the side effect of taste dysregulation, as the current P2X3-based cough-suppressant. These findings offer a druggable allosteric modulation strategy for treating chronic cough.

## Methods
The study was approved by the ethics committee of the China Pharmaceutical University.

### Drugs, mutagenesis, and cell cultures
Unless otherwise stated, all compounds were purchased from Sigma-Aldrich (U.S.). hP2X3 plasmid was purchased from Open Biosystems; cDNAs for hP2X1, hP2X2 and hP2X7 were synthesized by GKN and subcloned into pcDNA3. 1 vector; pcDNA3-rP2X2, pcDNA3-rP2X3 and pcDNA3-hP2X4 plasmids were generously gifted by Dr. Alan North and Dr. Linhua Jiang; the plasmid containing ANAP tRNA synthetase and the corresponding tRNAs was generously provided by Dr. Wei Yang. All mutations were generated with the KOD-Plus-Mutagenesis Kit (TOYOBO, catalog number KOD-401) and confirmed by DNA sequencing. Primers used to generate point mutations are summarized in Supplementary Table 2.

HEK293 (Catalog number: GNHu 43) and HEK293T (Catalog number: SCSP-502) cells were purchased from Shanghai Institutes for Biological Sciences (National Collection of Authenticated Cell Cultures, China) and cultured in Dulbecco's Modified Eagle Medium, supplemented with 10% fetal bovine serum (FBS), 1% penicillin-streptomycin, and 1% GlutaMAX™ at 37°C in a humidified atmosphere of 5% $CO_2$ and 95% air. Plasmids were transfected into cells by calcium phosphate transfection or Hieff Trans Liposomal transfection reagent.

### Conventional and nystatin-perforated whole-cell recordings
As described previously[77], currents of hP2X3, hP2X4, and rP2X2/3 were recorded using nystatin (Sangon Biotech) perforated recordings to avoid the rundown in current during multiple dose applications of ATP. Nystatin (0.15 mg/mL) perforated intracellular solutions contained (in mM) 75 $K_2SO_4$, 55 $KCl_2$, 5 $MgSO_4$, and 10 HEPES (pH 7.4). Currents of P2X2 and P2X7 receptors were recorded using a conventional whole-cell patch configuration. After 24–48 h of transfection, HEK293 cells were recorded at room temperature (25 ± 2°C) using an Axopatch 200B amplifier (Molecular Devices, USA) with a holding potential of -60 mV. Current data were sampled at 10 kHz, filtered at 2 kHz, and analyzed using PCLAMP 10 (Molecular Devices, USA) for analysis. HEK293 cells were bathed in standard extracellular solution containing (in mM) 2 $CaCl_2$, 1 $MgCl_2$, 10 HEPES, 150 NaCl, 5 KCl, and 10 glucose (pH 7.4, Tris-base). For conventional whole-cell recordings, the pipette solutions consisted of (in mM) 120 KCl, 30 NaCl, 0.5 $CaCl_2$, 1 $MgCl_2$, 10 HEPES, and 5 EGTA (pH 7.4, Tris-base). ATP and other drugs were dissolved in SS and applied to Y-tubes. N-phenylmaleimide (NPM), dithiothreitol (DTT), and 5,5'-dithiobis-(2-nitrobenzoic acid) (DTNB) were diluted to the appropriate concentrations and perfused into the cell membrane immediately during the recording period.

### Protein purification and microscale thermophoresis (MST)
A DNA fragment of the extracellular domain of the human P2X3 receptor (hP2X3-ECD, Glu46 – Ile318, as shown in Supplementary Fig. 5) was amplified by PCR with a 6xHis-Twin Strep-SUMO tag at the N-terminal. The obtained DNA fragment was ligated to the vector pET-28a (Novagen, Darmstadt, Germany). The ligated vector was transformed and expressed in *E. coli* Rosetta Origami2 strain (Shanghai Weidi Biotechnology Co., Ltd., China). Bacteria were cultured in Luria-

Bertani (LB) medium and overexpression was induced with 0.5 mM isopropyl b-D-1-thiogalactoside at 20 °C for 16 h. After overexpression and sonication, the mixture was centrifuged at 15,000 × *g* for 40 min. The supernatant was added to TALON metal affinity resin (Acara Biomedical Technology (Beijing) Co., Ltd., China) and gradually eluted with 20 mM HEPES buffer (pH 7.5) containing 500 mM NaCl and 300 mM imidazole. The eluate was loaded onto a Superdex® 200 Increase 10/300 GL column (Cytiva, Danaher Corporation, USA) for further purification. The proteins were confirmed using 15% sodium dodecyl sulfate (SDS)-polyacrylamide gel electrophoresis (PAGE).

The purified recombinant proteins were added to a buffer solution containing 20 mM HEPES (pH 7.5) and 500 mM NaCl and labeled according to the protocol of the protein labeling kit RED-NHS (Nanotemper, cat. no. L001). All tested stock compounds (10 mM) dissolved in dimethyl sulfoxide (DMSO) were diluted in the same buffer for the final MST assay. MST was performed using a Monolith NT.115 instrument (Nano Temper Technologies). The labeled proteins (2 µM) were mixed with the indicated concentrations of candidate compounds in a reaction buffer containing 0.05% Tween-20. MST data were then collected using Monolith NT.115 instrument (NanoTemper Technologies), and the dissociation constant was determined by Nanotemper analysis software (v2.2.4).

### Chemical synthesis of PSFL2915
Quercetin and other screened chemicals (Supplementary Fig. 3a) were purchased from TOPSCIENCE™ Co., Ltd, China (Tao-Shu). PSFL2915 was synthesized as described previously with slight changes[78]. Briefly, quercetin (7.23 g) was added to sulfuric acid (98%, 32 mL) with strong stirring. The reaction mixture was then stirred at room temperature for 96 h. The reaction mixture was diluted with 200 mL of water and incubated overnight. Subsequently, the reaction mixture was filtered and washed three times with saturated brine (30 mL) followed by one wash with ethanol (15 mL). The crude product was recrystallized in water three times, filtered, and dried to give the target product (yellow-green solid, yield 60%, Supplementary Fig. 6a). The molecular formula of PSFL2915 was assigned as $C_{15}H_{10}O_{13}S_2$ (molecular weight 462.35) determined from high-resolution mass spectral at m/z 229.97007 [M-2H]$^{2-}$ (theoretical m/z 229.97032 [M-2H]$^{2-}$) (Supplementary Fig. 6b). $^1$H NMR (300 MHz, DMSO-d6) δ 12.81 (s, $^1$H), 12.48 (s, $^1$H), 11.13 (s, $^1$H), 9.65 (s, $^1$H), 8.94 (s, $^1$H), 8.30 (d, *J* = 2.3 Hz, $^1$H), 7.93 (d, *J* = 2.3 Hz, $^1$H), 6.17 (s, $^1$H) (Supplementary Fig. 6c). $^{13}$C NMR (101 MHz, DMSO-d6) δ 176.12, 161.19, 160.86, 153.19, 147.27, 145.99, 144.77, 136.83, 131.02, 121.64, 119.87, 116.70, 110.08, 103.64, 98.60 (Supplementary Fig. 6d).

### Simulations and high-throughput virtual screening of hP2X3
Using the crystal structure of hP2X3 (PDB ID: 6AH4, 3.296 Å, ref. [79]) as a template, homology model of rP2X2 was constructed with MODELLER[80]; the quality of the models was checked with ProCheck[81]. The simulation systems were built using Maestro's System Builder, and conventional Molecular Dynamics (CMD) simulations were performed using DESMOND[82], as implemented in Maestro molecular modeling suite[83], using NPT (constant number (N), pressure (P), and temperature (T)) ensemble configurations, as we previously described[5,31,77]. A large 1-palmitoyl-2-oleoyl-sn-glycero-3-phosphocholine (POPC, 300 K) bilayer was chosen as the membrane system. The position of the membrane was determined based on the hP2X3 crystal structures (PDB IDs: 5SVJ and 5SVK) in the Orientations of Proteins in Membranes database (https://opm.phar.umich.edu)[84], and the automatic mode was selected to add the membrane. The ligand/hP2X3 complex was dissolved in simple point charge (SPC) water molecules. The solution conditions were neutral and counterions were added to compensate for the net negative charge of the system. In addition, the concentration of NaCl was chosen to be 150 mM. As we described previously, the DESMOND default relaxation protocol was applied to each system prior to the simulation run. Briefly[5,77,85], (1) simulations were performed

for 100 ps in the NVT (constant number (N), volume (V) and temperature (T)) ensemble with Brownian dynamics at 10 K and solute heavy atoms constrained; (2) simulations were performed for 12 ps in the NVT ensemble using a Berendsen thermostat at 10 K with small time steps and solute heavy atoms constrained[86]. (3) 12 ps simulations in the NPT ensemble using a Berendsen thermostat and Barostat at 10 K and 1 atm with solute heavy atoms constrained[85]. (4) 12 ps simulations using a Berendsen thermostat and Barostat at 300 K and 1 atm, with solute heavy atoms constrained; (5) 24 ps simulations using a Berendsen thermostat and Barostat at 300 K and 1 atm, with no constraints. After equilibration, the MD simulations were carried out for about 0.3–1.5 µs. The long-range electrostatic interactions were calculated using the smooth particle grid Ewald method[87]. The trajectory recording interval was set to 200-ps and the other default parameters of DESMOND were used in the MD simulation runs. All simulations used the all-atom OPLS_2005 force field[88,89], which is used for proteins, ions, lipids, and SPC waters. The Simulation Interaction Diagram (SID) module in Maestro[83] was used to explore the interaction analysis between ATP/quercetin and P2X3.

The default relaxation protocol of DESMOND was applied to each system before metadynamics (MetaD) simulations were run (same steps as for CMD simulations, see above). The parameters for MetaD simulations: (1) Width (rms width) of the repulsive Gaussian potential, in angstrom or degrees. Values are 0.05 Å for distances and 5° for dihedrals. (2) Height of the repulsive Gaussian potential (kcal/mol). (3) Time interval at which the repulsive Gaussian potential is added. All metadynamics simulations were run for 80-ns until they showed free diffusion along the defined collective variables (CV). Gaussian sums and free energy surfaces were generated by Maestro's metadynamics analysis tool. The bias $V(s, t)$ is usually constructed as a periodically increasing repulsive Gaussian, where s is the chosen CV and may be multidimensional. Thus, the free energy surface (FES) can be constructed in the space spanned by these CVs. The bias potential $V(s, t)$ at time $t$ can be written as:[90]

$$V(s,t) = \int_0^t \omega \exp\left(-\sum_{i=1}^d \frac{(S_i - S_i(t'))^2}{2\sigma_i^2}\right) dt' \qquad (1)$$

where $\omega$ is the Gaussian height controlled by the deposition stride, $S_i$ is one of $d$ CVs, and $\sigma i$ is the Gaussian width. This approach pushes the system to get rid of local minima and find the nearest saddle point on the FES. When a transient occurs, the deviation provides an estimate of the free energy as:

$$V(s,t) = -F(S) + C \qquad (2)$$

where C is an arbitrary additional constant. Since the absolute free energy is usually not important, this constant can be easily eliminated in the calculation of the free energy difference.

The virtual screening workflow (VSW) of the Schrödinger's Maestro suite[82] was used in the high-throughput virtual screening of the TARGETMOL database, using the default parameters we described before[91]. However, since we only screened a small library consisting of 20,000-30,000 compounds here, the XP mode was used directly.

All simulations were performed on a DELL T7920 with NVIDIA TESLTA K40C or CAOWEI 4028GR with NVIDIA TESLTA K80. The simulation system was prepared, trajectory analyzed, and visualized on a CORE DELL T7500 graphic workstation with 12 CPUs.

### Voltage-clamp fluorometry (VCF) analysis
As we previously described[91], HEK293T cells were cultured in a medium supplemented with 20 µm L-ANAP (AsisChem Inc.), and the corresponding tRNA, plasmids containing ANAP tRNA synthetase, and channel plasmids carrying wild-type (WT) or mutated genes were co-transfected into cells using Lipofectamine® 3000 reagent. The culture

medium (with L-ANAP) was changed every 8 h for 24 h and then replaced with an L-ANAP-free culture medium for 24 h. Fluorescence of ANAP was excited with a wLS LED light source (Photometrics) using BP340-390 excitation filters, DM410 dichroic filters and BA420-IF emission filters. The emission spectra of ANAP were captured using an Acton SpectraPro SP-2150 spectrometer (Princeton Instruments) and a Prime 95 B CCD camera (Photometrics), and the emission peaks were determined by fitting the spectra using a tilted Gaussian distribution. Briefly, the emission spectra of individual cells before and after administration of the agonist, ATP, were recorded with the use of an excitation filter of 340–390 nm (UV). For each cell, the maximum intensity was determined after subtracting the background, and the emission intensities at individual wavelengths were normalized to the maximum intensity. After fitting with the Gaussian equation, the peak emission wavelength was determined as the wavelength corresponding to the peak of the Gaussian curve. A shift in the peak emission wavelength (in nm) before and after agonist or antagonist administration reflects a conformational change at and near the ANAP-incorporated site induced by the drug and was used for quantification.

### Gel analysis
For engineered disulfide bond mutants, western blotting was performed as described previously[77]. Briefly, HEK293 cells were transfected with hP2X3-WT or mutant plasmids, washed three times with phosphate buffer (PBS; pH 7.4), and then RIPA lysate (200 µL) was added. Cells were collected from the bottom of the culture dish with a cell spatula and then centrifuged at 13,800 × g for 30 min at 4°C. Subsequently, a 20% (v/v) supernatant of each mutation was added twice to SDS loading buffer as protein samples, with (for reduction experiments) and without (for non-reduction experiments) β-mercaptoethanol (β-ME), followed by a 5 min metal bath. The protein samples were transferred to polyvinylidene difluoride (PVDF) membranes (Immobilon-P, Darmstadt, Germany). The membranes were blocked with 5% skim milk (Difco, Sparks, USA) for 2 h at room temperature. Anti-MYC (1:3000; catalog number 30601ES60, Yeason, China) and anti-GAPDH (1:3000; catalog number 60004-1-Ig, Proteintech Group, China) antibodies were incubated overnight at 4°C. After washing, the antibodies were incubated with secondary goat anti-mouse IgG(H + L)HRP (1:3000; catalog number LK2003L, Sungene Biotech, China) for 2 h at room temperature. Finally, the development procedure was performed as above.

### Cough suppression
**Animals.** Male Hartley guinea pigs (6-7 weeks) were purchased from the Nanjing Laifu (China). Male C57BL/6 WT mice (6-7 weeks) and male Sprague Dawley rats (6-7 weeks) were purchased from Jiangsu Huachuang Sino (China). P2X3 receptor knockout (*P2rX3*$^{-/-}$) mice were obtained using CRISPR/Cas9 technology and crossed back to the C57BL/6 strain for more than ten generations. All animals were maintained in a controlled environment (23 ± 2°C, 50 ± 10% humidity, 12 h light/dark cycle) with access to standard food and water. Consistent with previous findings (see below), only males were used in the experiment. All procedures were carried out in accordance with the Guide for the Care and Use of Laboratory Animals and approved by the ethics committee of the China Pharmaceutical University (2021-10-008).

**Drug doses.** In cough suppression assay, the doses of quercetin and gefapixant were calculated using the body surface area method[56] based on human doses, and the dose of PSFL2915 was the same as that of gefapixant, to facilitate comparison of efficacy. Animal equivalent dose (ADE) can be calculated with slight modification as: AED (mg/kg) = human dose (mg/kg) × (Km for human/Km for animal), where Km is the correction factor[56]. In dietary supplements, the daily dose of quercetin condensate recommended for human use is 500–2400 mg[55], and the

equivalent dose in mice is 103-492 mg/kg. Considering the solubility of quercetin and the maximum volume of mice, the doses of quercetin were determined to be 100 and 300 mg/kg/day twice daily (50 and 150 mg/kg, b.i.d.). The lowest clinically effective dose of gefapixant is 45 mg/kg (b.i.d.)[12], and the equivalent dose for mice is 9.25 mg/kg (b.i.d.). To ensure the efficacy of gefapixant in animal studies, the dose of gefapixant was determined to be 15 mg/kg (b.i.d.), and the doses of quercetin and PSFL2915 were the same as their effective doses in the cough test. The doses for guinea pigs were calculated in the same way. In the two-bottle taste preference experiment of mice, to show the advantage of quercetin and PSFL2915 in taste preference, the dose of gefapixant was only 1/3 of its effective dose in the cough test, which was calculated from the dose of taste-related adverse events of gefapixant in the phase 3 trial[12]. In a two-bottle taste preference assay in rats, gefapixant was administered at doses of 10 and 20 mg/kg, with reference to the previously described[57], and the doses of quercetin (200 and 400 mg/kg) and PSFL2915 (10 and 20 mg/kg) were calculated based on the ratio of their dosage to gefapixant in cough measurements.

**Cough measurements in awake guinea pigs.** Guinea pig coughs were measured as previously described, with slight modifications[57,92]. Male Hartley guinea pigs were placed in a glass cylinder and delivered 0.25 M citric acid aerosol and 10 µM ATP for 2 min (0.5 mL/min) using an ultrasonic nebulizer, followed by 9 min of observation. Cough sounds were recorded with a microphone placed in a glass jar, and cough frequency was recorded by visual observation of the animals. Guinea pigs were administered quercetin (25 or 100 mg/kg, b.i.d.), PSFL2915 (10 mg/kg, b.i.d.), gefapixant (as a positive control, 10 mg/kg, b.i.d., synthesized according to our previous description[31]), or 0.5% sodium carboxymethylcellulose solution (vehicle control) twice daily for 7 days. The number of coughs was measured 30 minutes after the last dose.

**Cough measurements in C57BL/6 *P2rX3*[+/+] and *P2rX3*[-/-] mice.** Mouse cough was examined as previously described with slight modifications[93,94]. Male C57BL/6 *P2rX3*[+/+] or *P2rX3*[-/-] mice were placed in a glass jar and supplied with 0.1% ammonia aerosol using an ultrasonic nebulizer and then observed for 5 minutes. The number of coughs was measured in the same way as in the guinea pig experiment. The differences in coughing between C57BL/6 WT mice and *P2rX3*[-/-] mice were compared without any intervention. In other experiments, animals were given oral doses of quercetin, PSFL2915, gefapixant or 0.5% sodium carboxymethylcellulose solution (vehicle control) twice daily for 7 days. The specific doses (b.i.d.) are shown in Fig. 9.

**Two-bottle taste preference in mice.** Two behavioral approaches were used to determine the effects of various drug doses on taste perception in male C57BL/6 mice. Animals were housed individually in ventilated cages and provided with two 50 ml bottles with lick tubes placed at equal height. To acclimatize the mice to the experimental environment, the mice were allowed to acclimatize to the two bottles containing water for three days. To study taste preference using aversive tastants, 0.3 mM of quinine was provided in one bottle. To stimulate drinking, mice were deprived of water the night before the test. Mice were administered quercetin (150 mg/kg, b.i.d.), PSFL2915 (15 mg/kg, b.i.d.), gefapixant (5 mg/kg, b.i.d., positive control), or 0.5% sodium carboxymethylcellulose solution (vehicle control) twice daily for 7 days. After the last dose, the mice were returned to their cages for 30 minutes, and then the two-bottle test was started. Again, to perform the same test, one bottle was replaced with 0.5 mM of acesulfame-K, a pleasant agent. The test was conducted for 48 hours and the bottles were replaced after 24 hours. The mice were not deprived of water overnight before the test drug was given throughout the test. Other experimental procedures were the same as for the aversive tasting

experiment, except that the drug was administered for 10 consecutive days.

**Two-bottle taste preference in rats.** A rat model for determining effects on taste perception was determined as previously described[57]. Male Sprague Dawley rats were housed individually in ventilated cages and provided with two 50-mL bottles with lick tubes placed at equal heights. To acclimatize the rats to the experimental environment, two bottles containing water were presented to the animals simultaneously for 15 min. To study taste preference with aversive tastants, 0.3 mM quinine was provided in one bottle for 15 min. The amount of water consumed in each bottle was measured. To stimulate water consumption, rats were deprived of water the night before the test. Different doses of gefapixant (0, 10, and 20 mg/kg), quercetin (0, 200, and 400 mg/kg), and PSFL2915 (0, 10, and 20 mg/kg) were administered intraperitoneally to each rat daily in increasing order. After injection, the rats were placed back in their cages for 45, 25, and 15 minutes (corresponding to the maximum plasma concentration of each compound) before the start of the double-bottle test.

## Data analysis

All electrophysiological recordings were analyzed using Clampfit 10.6 (Molecular Devices). Pooled data were expressed as mean and standard error r of mean (SEM). For comparisons between two groups, paired or unpaired tests were used, as appropriate. Comparisons between multiple independent groups were performed using one-way ANOVA followed by Dunnett's tests (Graphpad prism 8), unless otherwise stated. The concentration-effect curves were fitted using the Hill 1 equation: $I/I_{max} = 1/(1 + (IC_{50}/[inhibitor])^n)$, where $I$ is the normalized current at a given antagonist concentration, $I_{max}$ is the maximum normalized current induced by the agonist, $IC_{50}$ is the concentration of the antagonist exhibiting the half-maximum effect, *[inhibitor]* is the concentration of quercetin or PSFL2915, and $n$ is the Hill 1 coefficient.

## Reporting summary

Further information on research design is available in the Nature Portfolio Reporting Summary linked to this article.

## Data availability

The complete data generated in this study are described and provided in this document and its associated Supplementary Information and the Source data file. Previously solved structures were accessed from the PDB with accession codes: 5SVJ, 5SVK, and 6AH4. The MD simulation data have been deposited to figshare [https://doi.org/10.6084/m9.figshare.23963622]. All other data are available from the corresponding author upon request. Source data are provided with this paper.

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

## Acknowledgements

This work was supported by the Natural Science Foundation of Jiangsu Province (BK20202002 to Y.Y.), National Natural Science Foundation of China (31971146 and 32371289 to Y.Y.), Hunan Huxiang High-level Talent Program (2021RC5013 to Y.Y.), Xing Yao Leading Scholars of China Pharmaceutical University (2021) to Y.Y., Postgraduate Research & Practice Innovation Program of Jiangsu Province to X.Z. and Y.Y., CAMS Innovation Fund for Medical Sciences (2019-I2M-5-074 to Y.Y.), Medical Innovation and Development Project of Lanzhou University (lzuyxcx-2022-156 to Y.Y.), and State Key Laboratory of Utilization of Woody Oil Resource (2019XK2002 to C.L.).

## Author contributions

Y.Y. designed the project; Z.Z., X.Z., T.L., C.Y., and Y.G. performed cell culture, patch-clamp recording, and western blotting; M.S. performed VCF; Y.-H.G. performed MST; C.G., Z.Z., and X.Z. performed animal experiments; C. H. did the chimera hP2X4$^{P2X3(C107-C153)}$; Y-Y. L. performed patch-clamp recording; Y.Y. and P.C. did MetaD and MD simulations; Y.-T.L. and Q.L. performed the chemical synthesis of PSFL2915; Y.Y., C.G., and C.L. analyzed data; M.R. supervised Z.Z.; and Y.Y., M.Z., C.G., and W.W. wrote the manuscript. All authors discussed the results and commented on the manuscript.

## Competing interests

The authors declare no competing interests.
