## [Peer Review File · Nature Communications]

Reviewer #1

This is a very extensive study by C-R Guo Z-Z Zhang, X. Zhou and Collaborators focusing on specific structural features and allosteric modulation of the human P2X3 channel by Quercetin and a newly developed derivative PSFL2915. The compounds, which have chronic cough relieving properties, are shown to bind to the Inner Pocket of the Head Domain (IP-HD) of P2X3 receptor preventing the close-open conformational transition induced by ATP.

The work also examines the cough-suppressant properties of Quercetin and PSFL2915 in a mouse model of chronic cough, reporting a significant decrease in the frequency of coughing. This beneficial effect was absent in P2X3^{-/-} mice, suggesting that quercetin and PSFL2915 act on these channels. Similar effects were observed in a second animal model (Hartley guinea pig).

A behavioral study that completes this work suggests that Quercetin and PSFL2915 do not cause taste disturbance, a side effect associated with the administration of Gefapixant P2X3 receptor antagonist for the potential treatment of chronic cough.

The manuscript is clear and well-written and combines an impressive array of experimental and computational approaches to define the effects and mode of action of quercetin and PSFL2915 while also advancing knowledge on P2X channel structure-function relationship, gating and pharmacology.

Specific comments on Voltage Clamp Fluorometry:

Main points:

1) VCF experiments attempt to demonstrate conformational rearrangements occurring at the IP-HD domain, consequential to ATP binding. It is unclear how the superimposed ANAP emission spectra shown in Fig 2D were normalized. Please explain.

2) In Fig 2C the number of observations for WT (control) is surprisingly low (n=4) considering that the specificity and relevance of the ATP-induced left spectral shift of all fluorescent constructs depends on the behavior of WT. More importantly, the choice of P2X3-WT as "Control" is not ideal in this case, since no ANAP is incorporated. A better "negative control" should be a construct that incorporates ANAP in a region predicted not to undergo structural changes.

3) lines 163-166: Y37A mutation is introduced to delay desensitization and improve VCF signal.

Please explain how desensitization is expected to affect VCF signal (e.g. does P2X3 desensitization involve the same region?)

Minor:

Fig 3C: Note that the peak inward current for hP2X3G131C/E111C (middle trace) is cut.

Reviewer #2

The authors use an array of elegant approaches to identify a novel allosteric binding site on the hP2X3R subunit. Screening a small (<5000 compounds) chemical library identified the natural product quercetin as a modulator of P2X3 function working through this newly described allosteric binding site and synthesized an analog of quercetin with higher affinity and with in vivo efficacy in cough assays performed in guinea pigs and in humans.

Comments

- 1) It has been a common challenge when studying receptor and ion channel pharmacology that variations in structure have limited translation of results generated in rats or mice to other species, especially humans (PMID: 28473457; PMID: 17395593; PMID: 23709225; PMID: 9756382; PMID: 22639984). What is it about the novel structure targeted here that has facilitated translation from a human expression system to functional studies performed in both mice and guinea pigs?
- 2) The authors have overstated the literature supportive of antitussive actions of quercetin. There is in fact No supportive evidence, only circumstantial evidence for disease modification in conditions associated with cough. Corticosteroids and antibiotics are not antitussive but instead disease modifying. Prior to the present study the most logical interpretation of data generated using quercetin would be more consistent with disease modification.
- 3) Would the authors consider commenting on the surprising loss of function in taxifolin, differing from quercetin by a single carbon bond?
- 4) Consider adding a supplemental figure revealing the structure and perhaps in vitro pharmacological properties of gefapixant and other known ligands under clinical development targeting P2X3 receptors.
- 5) The taste aversion data are not impressive and yet some of the more provocative assertions about this new pharmacological target rely on these results. There is little in vitro evidence presented to

suggest that compounds structurally related to quercetin would lack the gustatory side effects of gefapixant, given the essential equipotency of quercetin and the synthetic analog described against homomeric P2X3 and heteromeric P2X2/3 receptors. More compelling results from taste aversion testing using a different assay have been published (PMID: 30902655). The authors efforts to explain these surprising results are speculative at best.

Reviewer #3

This is a very interesting and compelling manuscript where the authors identify a dietary supplement that acts as an allosteric modulator of P2X3 receptor (P2X3R) and strategically modify it to improve affinity to P2X3R. The authors claim that this antagonist binds at a different allosteric site than Gefapixant, a known P2X3R antagonist. They very nicely demonstrate that their novel antagonist does not have the same side effect profile (namely adverse effects on taste) as Gefapixant.

What I really enjoyed about this manuscript is the wonderfully broad and diverse types of experiments present – the authors span molecular modeling to molecular dynamics experiments to site-directed drug design to electrophysiology experiments to animal work! Fantastic.

A couple of general overall comments:

- 1) The figures are too “busy” and too small, especially Figure 2, Figure 3, and Figure 5. I really could not see the data well at all.
- 2) I found it a little difficult to follow the flow of the manuscript at times. For example, the authors suggest the reader compare Figure 6f and g before introducing Figure 6 at all. When things like this happen, I feel it interferes with the flow of the manuscript.

Some scientific questions/comments:

- 1) For the superimposition of hP2X3R in the resting and open state structures used to produce Figure 1, how were the two structures aligned? Was the overall trimer aligned or was a given protomer aligned between the two states? Does this affect the changes in the size and shape of the IP-HD between the two states? I think the authors should describe exactly how the two structures were aligned and if the way the structures are aligned affects the size of the IP-HD (and their conclusions).
- 2) For the experiments in Figure 2, where flUAA is incorporated into the receptor, do the authors know whether incorporation of the label affects the affinity of the modified receptor for ATP (direct binding affinity would be best but even showing that the EC50 of activation is unchanged from WT)? The authors state, “Despite the presence of flUAA Anap at these positions, ATP still caused channel opening”. However, from Figure 2C, lower panel, it looks like there are definitely differences in the sizes of the

currents. Is this because the affinities for ATP are different? Does the presence of the label limit ATP access to the binding pocket? I think these experiments might need to be performed.

3) The shifts in emission Lambda max values in Figure 2 are very modest. I think the authors need to do a control by adding 10 microMolar CTP or GTP (nucleotides that do not activate P2X3R) to verify that the emission changes are from specific binding of ATP and the conformational change in the receptor as opposed to just a change in emission because of increased charge (which might change the dielectric of the buffer..etc).

4) For the disulfide bond experiments shown in Figure 3, the authors do a nice job showing that oxidizing conditions limit ATP-induced current. They conclude that this is because the disulfide bond prevents the movements necessary for channel opening. However, an alternative explanation is that disulfide bond formation prevents ATP binding or decreases ATP affinity so the current is smaller at any given concentration of ATP. The authors need to show that the affinity for ATP binding is unchanged when the disulfide bond is formed. Thus, ATP binding still occurs but the conformational change cannot occur because of the disulfide bond. I think the same thing can be said for the DTNB experiments and the zinc ion bridge experiments.

5) Did the authors ever show that Gefapixant also blocks the ATP-induced fluorescence changes in flUAA Anap? The authors state that incorporation of flUAA Anap into sites inside the IP-HD would "most likely interfere with PSFL2915 binding". Were these experiments done? Does it actually interfere with PSFL2915 binding? If Gefapixant binds to a different site than PSFL2915 (outside the IP-HD), can gefapixant be used to block the emission shifts shown in Figure 2?

Reviewer #4

The authors present a generally very well written and logically flowing study where they have used state-of-the-art molecular biology, cell biology, chemistry, biochemistry etc methods and also animal models. The topic is very interesting and significant in that it reports a natural compound quercetine and its new, more potent derivative as cough suppressants acting via the P2X3 receptor in an allosteric manner. The derivative PSFL2925 shows equipotent inhibition of the P2X3 cation channel as gefapixant, a commercial drug. However, quercetine and PSFL2925 do not show taste disturbing effects in animals like the commercial compound does. The figures are highly informative and generally of high quality. The language is very good. I recommend the paper to be published after minor revisions. I have provided my detailed comments and suggestions in the annotated manuscript pdf file.

line 67. What is their mechanism of action/binding site?

Line 73. Pls refer here to the Figure where one can see these domains.

Line 85. Please check the order of the figures...should not S1 come before S2?

Line 117. Is this a result from MD simulations or just an observation when comparing the two structures, please stat this clearly...if an observation, the tempus could be present, not past: e.g. "based on the structural comparison of the resting and open states, the volume of...seems to undergo....."decreases from ...mainly results....

Line 120. Is this the same as "resting"? Or is it the open state without ATP?

Line 128. The random conformational selection of a unique 'snapshot' structure...?

Line 145. Why? what is the idea behind this method... what is CUA?

Line 151. How about 134? see next page...line 170

Line 213. What is the unit here?

Line 220. Rations of what? currents?

Line 284. How did you choose these CV?

Line 288. This was not mutated? see below

Line 293. S110 was not listed above?

Line 303. Do you have any idea why so? Is there some allosteric effect?

Line 307. Why did you choose this residue?

Line 398. Why did you use a smaller amount here? Did you also try with the full dose?

Line 427. Pls check the use of terms.

Line 436. How high dose of quercetin can/should be used? Maybe there are other side effects with higher doses?

Line 520. normal mode analysis? this was not mentioned in the study...

Line 541. What is RS?

Line 586. Where is the Kd seen in the figure?

Line 635. With vehicle?

Line 662. Why are the some of the residue labels in green? Please define

Line 665. Is this for the open state?

Line 674. Is it possible to make similar (comparable) figures? The view points seem to be so different that it is difficult to compare them...

Line682. Selected virtual hit compounds? Or already the experimentally tested hits?

Line 682. Selected virtual hit compounds? Or already the experimentally tested hits?

Line 702. ionic interactions are with negatively charged amino acids, which cannot directly interact with quercetin as it does not have any positive functional group...please make it clear that the interactions is with/through the Mg²⁺ cation...

Line 712. What is the value range on the y-axis?

Line 715. Contacts by the particular residue?

Line 717. Pls define the red dashed interaction lines

Line 791. This was not very clearly described in the results? Is this a previous result that has been published already?

Line 792. Did you use many structures? Please give then the PDB IDs for all. Please add also their resolution and the reference to the articles that report the structures.

Line 793. This reference is wrong...Caver is not Modeller...Sali lab has created the Modeller tool.

How many models were created, what was the modeling alignment (give for example in the Supplementary data?), which of the models did you choose for further investigations and what was the criteria? Stereochemical quality Any comment on the quality in the results?

Line 794. Simulations were carried out using DESMOND (ref), as implemented in Maestro molecular modeling suite (Schrödinger, LLC) (ref). The simulation systems were constructed employing Maestro's System Builder function....Please see Schrödinger website for how to cite their software....software versions should also be given, which version of Maestro was used...etc. Protein preparation wizard/workflow was also apparently used to add the hydrogens etc...?

Line 798 . The Orientations of Proteins in Membranes (OPM) database (pls give the ref).

Line 800 Na⁺ counter ions?

Line 805. References for all methods should be added ; Line 807. Ref ; Line 813. Ref. Line 811. Reference should be given; please make sure you did not use the U-series instead as this is the default in the more recent versions of Maestro/Desmond...

Line 815. Did you also study the PSFL2915, or only quercetin? would be interesting to know how the sulfogroups interact...

Line 819. which spacing?

Line 820. How do you observe this, what is the parameter to be followed?

Line 821. Please give the correct reference for the tool.

Line 824. This is likely not the original article describing the method? Please cite (also) the original?

Line 826. Pls check.... something is missing, and should be deleted 'd'?

Line 828. Transient what?

Line 834. Pls describe shortly how you screened, using Glide? Which mode? what was the criteria to select the hits for testing?

Point-to-point responses to reviewers' comments (*comments in italics*)

Reviewer #1 (Remarks to the Author)

This is a very extensive study by C-R Guo Z-Z Zhang, X. Zhou and Collaborators focusing on specific structural features and allosteric modulation of the human P2X3 channel by Quercetin and a newly developed derivative PSFL2915. The compounds, which have chronic cough relieving properties, are shown to bind to the Inner Pocket of the Head Domain (IP-HD) of P2X3 receptor preventing the close-open conformational transition induced by ATP.

The work also examines the cough-suppressant properties of Quercetin and PSFL2915 in a mouse model of chronic cough, reporting a significant decrease in the frequency of coughing. This beneficial effect was absent in P2rX3^{-/-} mice, suggesting that quercetin and PSFL2915 act on these channels. Similar effects were observed in a second animal model (Hartley guinea pig).

A behavioral study that completes this work suggests that Quercetin and PSFL2915 do not cause taste disturbance, a side effect associated with the administration of Gefapixant P2X3 receptor antagonist for the potential treatment of chronic cough.

The manuscript is clear and well-written and combines an impressive array of experimental and computational approaches to define the effects and mode of action of quercetin and PSFL2915 while also advancing knowledge on P2X channel structure-function relationship, gating and pharmacology.

Response: We are very grateful to the reviewer for his/her positive comments on the manuscript. We have added additional experiments and revised the manuscript according to the reviewer's suggestions.

Specific comments on Voltage Clamp Fluorometry:

Main points:

- 1. VCF experiments attempt to demonstrate conformational rearrangements occurring at the IP-*

HD domain, consequential to ATP binding. It is unclear how the superimposed ANAP emission spectra shown in Fig 2D were normalized. Please explain.

Response: We thank the reviewer for bringing up this point. We now describe the normalization of the VCF data in more detail in Methods section of the revised manuscript (lines 873 – 880, page 51) as: the emission spectra of individual cells before and after administration of the agonist, ATP, were recorded with the use of an excitation filter of 340–390 nm. For each cell, the maximum intensity was determined after subtracting the background, and the emission intensities at individual wavelengths were normalized to the maximum intensity. After fitting with the Gaussian equation, the peak emission wavelength was determined as the wavelength corresponding to the peak of the Gaussian curve. A shift in the peak emission wavelength (in nm) before and after agonist or antagonist administration reflects a conformational change at and near the ANAP incorporated site induced by the drug and was used for quantification.

Notably, two different protocols are currently used to detect conformational changes using the VCF method. One scheme uses the *Xenopus* oocyte expression system, with the site of interest substituted by a cysteine residue, which can be specifically labeled with fluorescent molecules such as tetramethyl-rhodamine-maleimide. The conformational changes in the labeled region are inferred from changes in fluorescence intensity emitted by the fluorescent probe upon agonist activation (Albert et al., *Nature*, 1999; Lorinczi et al., *PNAS*, 2012; Roberts et al., *J Neurosci*, 2007). Due to the low membrane permeability of the currently used fluorescent tools, this method is typically used to label the extracellular site of the protein (Kalstrup et al., *PNAS*, 2013; Talwar et al., *Neuropharmacology*, 2015). Moreover, the specificity of the fluorophore labeling is problematic because of the possible presence of free cysteine in not only other regions of the target protein but also other endogenous proteins on the oocyte membrane (Andriani et al., *Elife*, 2021).

The other scheme, as in our case, uses HEK293T cells, which have a relatively small membrane surface, express fewer receptors, and have weaker absolute fluorescence intensities compared to oocytes. Therefore, instead of inferring conformational changes based on changes in absolute fluorescence intensity, this scheme uses the principle that the emission spectrum of the unnatural amino acid ANAP changes in different environments. Thus, the amino acid of interest is mutated to incorporate ANAP using the UAG codon and engineered tRNA. Upon expression of the mutant protein, agonist-induced conformational change is inferred based on the shift of the ANAP peak emission wavelength (Lee, et al., *J Am Chem Soc*, 2009; Chatterjee et al., *J Am Chem Soc*, 2013). Compared to the oocyte system, this method has higher site specificity, fewer perturbations on the protein structure (Sakata et al., *PNAS*, 2016) and is more sensitive to slight conformational changes of the receptor (Kalstrup et al., *PNAS*, 2013). The conformational change of IP-HD of P2X3 was achieved by recording the shift of the peak emission wavelength before and after ATP administration.

2. *In Fig 2C the number of observations for WT (control) is surprisingly low (n = 4) considering that the specificity and relevance of the ATP-induced left spectral shift of all fluorescent constructs depends on the behavior of WT. More importantly, the choice of P2X3-WT as “Control” is not ideal in this case, since no ANAP is incorporated. A better “negative control” should be a construct that incorporate ANAP in a region predicted not to undergo structural changes.*

Response: We thank the reviewer for bringing up this important issue. Choosing correct controls is indeed critical when using the VCF method. WT cells were chosen as a negative control for two reasons: 1) ANAP can non-specifically aggregate in the cell, which gives rise to fluorescence signals that are largely insensitive to extracellular administration of compounds such as ATP; 2) these non-specific fluorescence signals in WT-expressing cells are equally affected by unavoidable factors, such as fluorescence quenching, as the signals produced by ANAP incorporated in the mutant P2X3 receptors.

As suggested by the reviewer, we also added an additional control with ANAP incorporated at V143 (V143^{ANAP}; Fig. R1a). This site is directly exposed to the solution and has few other amino acids nearby. According to the principle that contributes to the alteration of the ANAP emission (Lee et al., J Am Chem Soc, 2009), conformation changes at V143^{ANAP} are unlikely to shift its emission spectrum. As expected, the emission peak of V143^{ANAP} only fluctuated between -1 nm and 3 nm after the ATP treatment (P = 0.2215, paired t-test, n = 5; Fig. R1b, c). For WT (no ANAP insertion but non-specific intracellular accumulation), the 'ANAP' fluorescence was much weaker than emission peak did not differ before and after ATP administration (P = 0.8417, paired t-test, n = 5, Fig. R1c). Furthermore, there was no significant difference between 'noise' (the change value of WT) and V143^{ANAP} (P = 0.4583, unpaired t-test, n = 5-7, Fig. R1d). In addition, mutations such as E111^{ANAP} at the positive site still produced significant differences in the maximum absorption wavelength shift using V143^{ANAP} as a control (P < 0.05, one-way ANOVA with Dunnett's multiple comparison tests, F (10, 49) = 7.611, n = 3 - 12). Thus, changes in the key site of IP-HD were evident both in terms of systematic 'noise' (WT) and using a P2X3 mutant that does not undergo a significant wavelength shift (V143^{ANAP}) as a control, suggesting that conformational changes in IP-HD can indeed be detected by VCF.

Second, we increased the number of P2X3 WT repeats to n = 6 as suggested by the reviewer (new Fig. 2c).

The above data have been incorporated into the new Fig. 2 in the revised manuscript.

Fig. R1. ATP induces no change in the emission spectrum of V143^{ANAP}. (a) Location of V143 in the extracellular side of P2X3, exposed to the solution and outside of IP-HD. (b) Representative emission spectra of cells expressing V143^{ANAP} (black, bath solution; red, solution containing 10 μM ATP). (c-d) Summary of peak shifts in ANAP emission wavelengths in the absence and presence of ATP (n = 5-6, P > 0.05, ATP-treated vs. control, paired t-test (c) and p > 0.05 V143^{ANAP} vs. WT(d)).

3. lines 163-166: Y37A mutation is introduced to delay desensitization and improve VCF signal. Please explain how desensitization is expected to affect VCF signal (e.g. does P2X3 desensitization involve the same region?)

Response: We thank the reviewer for this question. Because VCF involves emission scanning, the sampling rate is much slower than electrophysiological recording. Thus, for fast desensitizing receptors like P2X3, it was possible that VCF might not capture the conformational change associated with channel opening. However, receptor desensitization does not necessarily occlude the acquisition by VCF of peak emission wavelength shift. For some rapidly desensitizing receptors, the inserted fluorophore may still be exposed to the changed environment and thus display altered fluorescence signals for a period of time after the current rundown before the receptor fully returns to the resting state. In such cases, VCF

can still be used to detect conformational changes, despite the extremely short duration of the open state. For example, Éva Lörinczi et al. studied the activation and desensitization of the P2X1 receptor using VCF and showed that the change in fluorescence intensity of the P2X1 receptor may continue to maintain its original value after desensitization (Lorinczi et al., PNAS, 2012).

However, to rule out the possibility that rapid receptor desensitization may hamper the detection of conformation changes by VCF, we introduced the delayed desensitization mutation Y37A (Giniatullin and Nistri, Front Cell Neurosci, 2013) into the hP2X3 receptor; With the introduction of Y37A, we observed a shift in the peak emission wavelength of hP2X3^{V37A}-E156^{ANAP}, which was not detected in hP2X3-E156^{ANAP}. Thus, by delaying desensitization, we increased the probability of observing the conformational change using VCF to examine the incorporated flUAA.

Unlike hP2X3-E156^{ANAP}, the introduction of Y37A did not alter the ability of VCF to detect conformational changes at other mutated sites within IP-HD (new Fig. 2c). None of these mutations resulted in a dramatic change from fast to slow desensitization of P2X3 (Fig. R2), suggesting that the conformational changes observed in VCF are associated with opening rather than desensitization.

Fig. R2. Typical traces of ATP currents of mutants within IP-HD (n = 3-5).

Minor:

4. Fig 3C: Note that the peak inward current for hP2X3G131C/E111C (middle trace) is cut.

Response: We are very sorry about this, and we have adjusted the figure to show the complete trace (new Fig. 3c).

Reviewer #2 (Remarks to the Author)

The authors use an array of elegant approaches to identify a novel allosteric binding site on the hP2X3R subunit. Screening a small (<5000 compounds) chemical library identified the natural product quercetin as a modulator of P2X3 function working through this newly described allosteric binding site and synthesized an analog of quercetin with higher affinity and with in vivo efficacy in cough assays performed in guinea pigs and in humans.

Response: We are very grateful to the reviewer for his/her many constructive suggestions and positive comments on our manuscript. In response to the reviewer's suggestions, we have performed additional experiments and simulations to strengthen our conclusions.

Comments

1. It has been a common challenge when studying receptor and ion channel pharmacology that variations in structure have limited translation of results generated in rats or mice to other species, especially humans (PMID: 28473457; PMID: 17395593; PMID: 23709225; PMID: 9756382; PMID: 22639984). What is it about the novel structure targeted here that has facilitated translation from a human expression system to functional studies performed in both mice and guinea pigs?

Response: We thank the reviewer for pointing out this important issue. The issue of species differences is a common challenge when studying receptors and ion channels, including P2X receptors. For example, compounds such as the P2X4 receptor inhibitor BX430 and the P2X7 receptor inhibitor GW791343 show significant species differences (Ase, Honson et al., *Mol Pharmacol*, 2015; Michel, Chambers et al., *Br J Pharmacol*, 2008). We also encountered this problem when studying the P2X7 receptor; we found tens to hundreds of fold differences in the IC₅₀ of inhibitors against human P2X7 and mouse P2X7.

However, species differences in P2X3 inhibitors are relatively rare; P2X3 inhibitors currently tested in clinical trials, such as Merck Sharp & Dohme's AF219/gefapixant, Belus' BLU5937, and Shionogi's S600918, do not differ significantly at the P2X3 receptor levels in humans and experimental animals. We have also evaluated the inhibitory effects of S600918 and its derivative DDTPA (differing only by few substituents) on P2X3 receptors in different species and found no significant differences in the inhibitory effects of DDTPA on rat, mouse and human P2X3 receptors (rP2X3, mP2X3 and hP2X3) (rP2X3_{IC50} = 0.304 ± 0.05 μM, mP2X3_{IC50} = 0.299 ± 0.08 μM and hP2X3_{IC50} = 0.170 ± 0.04 μM, n = 5; Fig. R3a). Similarly, we previously demonstrated that AF353, a derivative of Merck Sharp AF219/gefapixant (differing by only one group substituent), had almost completely overlapped concentration response curves for hP2X3 (IC₅₀ = 12.9 ± 0.5 nM) and rP2X3 (IC₅₀ = 10.2 ± 1.4 nM) (Wang J. PNAS, 2018; attached as Fig. R3b for reviewer reference).

Fig. R3. Concentration-response curves of P2X3 antagonists DDTPA (a) and AF353 (b) on P2X3 receptors of different species. The solid lines are fits to the Hill 1 equation; n = 3 - 6.

We are fully aware that difference in efficacy between model animals and humans is a constant topic in ion channel drug development, and we pay close attention and great efforts to address this issue. That is why we used three model animals, mice, rats, and guinea pigs (*cavia porcellus*) in this study to demonstrate *in vivo* efficacy. In cell-based assays, we tested effects of PSFL2915 on hP2X3, rP2X3, mP2X3 and *cavia porcellus* P2X3 (cpP2X3), respectively. Similar to S610098 and gefapixant and their analogs, the inhibitory effects of PSFL2915 on rP2X3, mP2X3 and hP2X3 were not significantly different (mP2X3-IC₅₀ = 0.276 ± 0.001 μM, rP2X3-IC₅₀ = 0.142 ± 0.012 μM, hP2X3-IC₅₀ = 0.319 ± 0.029 μM. Fig. R4). The data on mP2X3

and rP2X3 are importance because mice and rats are often used to test the drug's efficacy and side effects. The apparent affinity of PSFL2915 on cpP2X3 was slightly lower than that on hP2X3 (cpP2X3 $IC_{50} = 1.47 \pm 0.17 \mu\text{M}$, Fig. R4), which is a relatively small difference. Considering the superior value of the guinea pig cough model and the fact that PSFL2915 had a slightly higher apparent affinity for hP2X3 than cpP2X3, we view guinea pigs as being appropriate for testing the efficacy since the cough suppressant effect of this drug should be even stronger in humans than in guinea pigs.

Fig. R4. Concentration-response curves of PSFL2915 at hP2X3, rP2X3, mP2X3 and cpP2X3 receptors. The solid lines are fits to the Hill 1 equation; $n = 3 - 5$.

2. The authors have overstated the literature supportive of antitussive actions of quercetin. There is in fact No supportive evidence, only circumstantial evidence for disease modification in conditions associated with cough. Corticosteroids and antibiotics are not antitussive but instead disease modifying. Prior to the present study the most logical interpretation of data generated using quercetin would be more consistent with disease modification.

Response: We thank the reviewer for mentioning this point. To the best of our current knowledge, quercetin has been described to be able to ameliorate cough (Wang et al., *J Anal Methods Chem*, 2015; Zhu et al., *Chem Biodivers*, 2018; Zupanets et al., *Zaporozhye Medical Journal*, 2021). For example, Zhu et al. suggested that both quercetin and herbal extracts containing quercetin were able to reduce the number of coughs induced by ammonia in mice (Zhu et al., *Chem Biodivers*, 2018); Zupanets et al. demonstrated that quercetin improved cough symptoms in patients with COVID-19 (Zupanets et al., *Zaporozhye Medical Journal*,

2021). In addition, quercetin is an indicator component chosen for the development of quality controls of many cough herbs (Wang et al., *J Anal Methods Chem*, 2015). In Europe and the United States, quercetin is also marketed by large health food companies such as GNC for improving the symptoms of pulmonary nodules and coughing and wheezing.

However, as the reviewer suggested, the improvement in cough symptoms is only one indicator of the observed effects, which may be a direct cough suppressant effect of the drug or an indirect effect of the drug by improving other pathways. Here, we demonstrate that quercetin directly suppresses cough in mice, which can be at least partially explained by its inhibition of P2X3 receptors (new Fig. 9). Nevertheless, to avoid unnecessary misunderstandings, we made appropriate adjustments in the description of the cough suppressant effect of quercetin as suggested by the reviewer (lines 45-47, page 3; line 356, page 17; lines 476-479, page 22, in the revised manuscript).

3. Would the authors consider commenting on the surprising loss of function in taxifolin, differing from quercetin by a single carbon bond?

Response: We thank the reviewer for raising this point. Taxifolin has one less double bond compared to quercetin, which leads to a more flexible heterocycle, resulting in a different conformation from quercetin (Fig. R5a). We have added a comparison of the hP2X3/taxifolin complex with the hP2X3/quercetin complex in conventional molecular dynamics (CMD) simulations on the μs time scale (Figs. R5, 6). The change in the conformation of the hexameric ring resulted in significantly more fluctuations of taxifolin than quercetin in the hP2X3/ligand complexes during the 2 μs CMD simulations (taxifolin RMSD (Fig. R6c) compared to quercetin RMSD (new Fig. S7b)). The rotatable bond fluctuations of taxifolin (Fig. R6b) were also increased compared to that of quercetin (new Fig. S7c), and the interactions with IP-HD were significantly altered (some quercetin-like interactions disappeared (Fig. R6d, lower dashed box), while others increased, Fig. R6d, upper dashed box). These indicate that the binding of taxifolin to IP-HD (Fig. R5c) is unstable. After contacting hP2X3, taxifolin needs

to be squeezed into the pocket between loops 108-118 and 122-134 (Fig. R5b, upper panel), which requires partial binding free energy to compensate for the increased receptor strain energy.

In contrast, the binding of quercetin remained stable even after more than 5 μ s of CMD simulations, and quercetin fit the shape of the IP-HD pocket well (Fig. R5b, lower panel). Secondly, during the CMD simulation, the flexibility of the hexameric ring of taxifolin increases and the protrusion of the C atom (Fig. R5) leads to a certain change in the angle of E109 to the phenolic hydroxyl group of taxifolin and the internal Mg^{2+} of IP-HD (Fig. R5c, d), while the rigidity of this position of quercetin is well suited for the interaction of E109 with Mg^{2+} (Fig. R5d). Supporting a key role of E109 in ligand recognition, the E109A mutant changed the apparent affinity of quercetin by more than 50-fold compared to the WT channel (new Fig. 6g).

Fig. R5. Conformational differences between taxifolin and quercetin (a) and comparison of hP2X3 in complexes with quercetin and taxifolin after CMD simulated receptor-ligand interactions for ~ 2 -5 μ s time scale (b-d).

Fig. R6. Conventional molecular dynamics simulations to study the interaction of hP2X3 with taxifolin. (a) The interaction between P2X3 and taxifolin was monitored throughout the MD simulations. hP2X3/taxifolin interactions were classified into four types: hydrogen bonds (green), hydrophobic (light purple), ionic (pink), and water bridges (blue). The stacked histograms are normalized over the course of the trajectory. (b) Backbone root-mean-square deviation (RMSD) analysis of the binding process of taxifolin to hP2X3 throughout the MD simulation (0-2000 ns). (c) Torsion plots of quercetin summarizing the conformational evolution of each rotatable bond every 10 ns throughout the simulated trajectory. The radial plots represent the conformation of the torsion bodies. Center of the radial plot represents the beginning of the simulation, plotting the temporal evolution in the radial direction outward. The histogram summarizes the data of the corresponding radial plot, which represents the probability density of the torsion. The relationship between histogram and torsional potential

gives insight into the conformational strain that the ligand underwent to maintain the hP2X3-bound conformation. (d) Time-dependent contacts between P2X3 and /taxifolin. The upper panel represents the total contacts of P2X3 with taxifolin; the lower panel represents the contacts of single amino acid with taxifolin. The lower panel represents the contacts of individual amino acids with taxifolin. The darker the orange color, the more number of contacts the residue has with the ligand, as some amino acids have multiple specific contacts with the ligand. Dashed rectangles indicate significantly altered interactions between the taxifolin and hP2X3 during simulations.

Thus, the removal of the double bond relative to quercetin caused a change in taxifolin conformation, which may lead to its low inhibition of hP2X3, even at taxifolin concentration up to 100 μ M.

4. Consider adding a supplemental figure revealing the structure and perhaps in vitro pharmacological properties of gefapixant and other known ligands under clinical development targeting P2X3 receptors.

Response: We thank the reviewer for her/his suggestions. We have added Supplementary Table 1 to the revised manuscript. Only the structure of some of the compounds currently in the clinical trials are disclosed, as follows:

Table 1 Structures and possible in vitro pharmacological properties of inhibitors targeting the P2X3 receptor

Compounds	Structure	hP2X3 IC ₅₀
S-600918 ^a		67.5 \pm 1.2 nM
Gefapixant ^b		76 nM ^d

BLU-5937^b

25 nM^e

Eliapixant^c

8 nM^f

Agonists:^a 1 μ M ATP, ^b 3 μ M α - β -methylene ATP, ^c 10 μ M α - β -methylene ATP.

References: ^d McGarvey, Birring et al. 2022. ^e Garceau and Chauret 2019. ^f Davenport, Neagoe et al. 2021.

5. *The taste aversion data are not impressive and yet some of the more provocative assertions about this new pharmacological target rely on these results. There is little in vitro evidence presented to suggest that compounds structurally related to quercetin would lack the gustatory side effects of gefapixant, given the essential equipotency of quercetin and the synthetic analog described against homomeric P2X3 and heteromeric P2X2/3 receptors. More compelling results from taste aversion testing using a different assay have been published (PMID: 30902655). The authors efforts to explain these surprising results are speculative at best.*

Response: We are very grateful to the reviewers for raising this important issue. As suggested by the reviewer, we first supplemented the data from the taste aversion experiment according to the protocol in PMID: 30902655. The results showed that the volume of quinine solution (bitter sensation) consumed by the rats increased significantly after intraperitoneal injection of 10 and 20 mg/kg of gefapixant ($P < 0.05$, paired t-test, $n = 10$, Fig. R7). In contrast, the volume of quinine solution consumed by the rats did not change significantly after intraperitoneal injection of 200 and 400 mg/kg of quercetin ($P > 0.05$, paired t-test, $n = 10$, Fig. R7). In addition, there was no significant change in the volume of quinine solution consumed by rats after intraperitoneal injection of 10 and 20 mg/kg PSFL2915 ($P > 0.05$, paired t-test, $n = 10$, Fig. R7). These results suggest that gefapixant causes taste impairment in rats, whereas quercetin and PSFL2915 do not significantly affect taste perception in rats (Fig. R7), which is consistent with the results of our original manuscript. These results have been incorporated into new Fig. 9 in the revised manuscript.

Fig. R7. Effects of gefapixant, quercetin, and PSFL2915 on taste perception in rats. n.s. $P > 0.05$, * $P < 0.05$ compared with vehicle; paired t-test; $n = 10$.

In addition, improving the selectivity of compounds between P2X3 homotrimers and P2X2/3 heterotrimers may be the most important point in reducing the side effects of P2X3 receptor inhibitors for taste disorders, but it may not be the only way. A recent work even suggested that the selectivity of P2X332 heterotrimers and P2X333 homotrimers is not necessarily a key factor in drug candidates triggering taste abnormalities in humans (High, Jetté et al., *ERJ Open Research*, 2023). The following factors may also have an impact on the drug's side effect of taste disorders:

First, a permeability barrier similar to the "blood-brain barrier" exists around the taste buds of the tongue epithelium, which prevents the penetration of many (but not all) compounds, and this barrier is effective for both compounds injected directly into the tongue and those that reach the taste buds through the blood circulation (Mistretta, *Am J Physiol*, 1971; Michlig et al., *J Comp Neurol*, 2007; Dando et al., *Am J Physiol Cell Physiol*, 2015). The presence of a permeability barrier around taste buds in the epithelium of the tongue is not only important for the design of taste modulators, but also for the lack of effect of many drugs on taste (Damak S, *Am J Physiol Cell Physiol*, 2015). The ability of different compounds to penetrate the peripalantine barrier varies greatly depending on their physicochemical properties, and the ability of a drug to cross the peripalantine barrier can be reduced by altering its structure, thereby reducing its effect on taste perception (Michlig, et al., *J Comp Neurol*, 2007; Damak, *Am J Physiol Cell Physiol*, 2015).

Second, there are differences in tissue distribution of administered drugs. The drug needs to reach a certain concentration in the target organ to be effective (including side effects); thus, the tissue distribution of the drug has an important effect on its efficacy and side effects. One study measured the distribution of quercetin in different tissues of rats after adding 0.1% or 1% of quercetin to the diet of rats for 11 weeks. The results showed that quercetin had the highest concentration in the lungs and the lungs contained the highest total amount of quercetin and its metabolites (Cai, Chen et al, Ai Zheng, 2004; de Boer, Dihal et al., J Nutr,2005). This may be one of the reasons why quercetin did not produce significant taste impairment.

Another reason for the lack of taste side effects of quercetin might be related to Mg^{2+} . Mg^{2+} plays an important role in the inhibition of P2X3 receptors by quercetin in that the degree of inhibition is positively correlated with the Mg^{2+} concentration (new Figs. 5d and S4). Mg^{2+} is released from inflamed tissues, which can make P2X3 receptors more sensitive to quercetin under inflamed conditions.

Our results that quercetin does not alter taste perception in rodents are also consistent with the experience of long-term use of quercetin in the population and with the results of clinical trials on quercetin. The safety of quercetin was demonstrated in a phase I clinical trial study: no significant adverse effects were observed when quercetin was administered at a dose of 5 g/day (Lu et al. Phytother Re, 2016). Since taste impairment is a category of adverse reactions easily perceived by patients, it should be reasonable to believe that quercetin may not have taste-related adverse reactions. In addition, several announced clinical trials have demonstrated the safety of quercetin at the dose of 1 g/day (Harwood et al., Food Chem Toxicol, 2007; Heinz et al., Pharmacol. Res., 2010).

Therefore, in addition to the selectivity for P2X3 homotrimers and P2X3/2 heterotrimers, factors such as the physicochemical properties and in vivo distribution of P2X3 receptor

inhibitors may also influence the taste dysregulation effects of these drugs, the mechanisms of which are complicated and require further investigation.

Reviewer #3 (Remarks to the Author)

This is a very interesting and compelling manuscript where the authors identify a dietary supplement that acts as an allosteric modulator of P2X3 receptor (P2X3R) and strategically modify it to improve affinity to P2X3R. The authors claim that this antagonist binds at a different allosteric site than Gefapixant, a known P2X3R antagonist. They very nicely demonstrate that their novel antagonist does not have the same side effect profile (namely adverse effects on taste) as Gefapixant.

What I really enjoyed about this manuscript is the wonderfully broad and diverse types of experiments present – the authors span molecular modeling to molecular dynamics experiments to site-directed drug design to electrophysiology experiments to animal work! Fantastic.

Response: The reviewer's kind words are greatly appreciated. We have added additional experiments and revised the manuscript according to his/her suggestions.

A couple of general overall comments:

1. The figures are too “busy” and too small, especially Figure 2, Figure 3, and Figure 5. I really could not see the data well at all.

Response: We are very grateful to the reviewer for the suggestion, which has greatly helped the readability of the manuscript. In the revised manuscript, we have reorganized the original figures 2, 3 and 5, which now spread over 10 new figures, as suggested by Reviewer #3 and Editors.

2. I found it a little difficult to follow the flow of the manuscript at times. For example, the authors suggest the reader compare Figure 6f and g before introducing Figure 6 at all. When things like this happen, I feel it interferes with the flow of the manuscript.

Response: We combed through the manuscript, corrected similar errors, and rearranged the relevant figures to improve its readability.

Some scientific questions/comments:

3. For the superimposition of hP2X3R in the resting and open state structures used to produce Figure 1, how were the two structures aligned? Was the overall trimer aligned or was a given protomer aligned between the two states? Does this affect the changes in the size and shape of the IP-HD between the two states? I think the authors should describe exactly how the two structures were aligned and if the way the structures are aligned affects the size of the IP-HD (and their conclusions).

Response: We thank the reviewer for bringing up this concern. For the P2X3 trimer, it is true that superposition cannot be performed with PYMOL. We used Schrödinger Suite's Protein Structure Alignment Module for the superposition of trimers (based on both sequence and 3D structure, not the forced alignment). This superposition does not change any atomic relative coordinate parameters or cause any undesirable shifts occur. Therefore, there is no stretching of the IP-HD and no change in the size of the IP-HD.

4. For the experiments in Figure 2, where fUAA is incorporated into the receptor, do the authors know whether incorporation of the label affects the affinity of the modified receptor for ATP (direct binding affinity would be best but even showing that the EC_{50} of activation is unchanged from WT)? The authors state, "Despite the presence of fUAA Anap at these positions, ATP still caused channel opening". However, from Figure 2C, lower panel, it looks like there are definitely differences in the sizes of the currents. Is this because the affinities for ATP are different? Does the presence of the label limit ATP access to the binding pocket? I think these experiments might need to be performed.

Response: We appreciate the reviewer for pointing out this concern. For the VCF assay, a stop codon (TAG) is used for incorporating the unnatural amino acid, ANAP, into the protein at the specific location by site-directed mutagenesis. The incorporation occurs when a plasmid

containing ANAP tRNA synthetase (AnapRS) is co-transfected with the modified cDNA for the target receptor into HEK293T or other cells. This *in vitro* inserted translation system is weaker than the endogenous translation system, resulting in typically low expression of the ANAP-incorporated proteins (Chatterjee et al., J Am Chem Soc, 2013). This is particularly true for receptors of long sequences, where low transfection efficiency, weak receptor expression, and small currents are common (Xu et al., Nat Commun, 2020). In our recent work (Sun et al., Sci Bull, 2022), we also found that when ANAP was inserted into TRPV1, the current size tended to be smaller than when ANAP was not included. It is generally believed that the current magnitude does not affect the conclusion of the VCF experiments since the interest is in the conformational change of a specific site. In fact, many of the published studies (e.g., Yang et al., J Gen Physiol, 2014; Yang et al., Nat Commun, 2018; Xu et al., Nat Commun, 2020; Yang et al., PNAS, 2020) did not even mention whether the ANAP incorporated channels still conducted any currents. Our purpose of showing channel currents in Fig.2 of the original manuscript was to show that the hP2X3 receptor remained functional after ANAP insertion and that the IP-HD conformational change was caused by ATP.

As in the case of any mutagenesis experiments, amino acid substitutions can alter the protein function. Therefore, to address the reviewer's concern, we measured the EC₅₀ of ATP for hP2X3^{E111ANAP}, a mutant displaying a significant change in the emission spectra of ANAP. Although the EC₅₀ was 2.93 ± 1.36 for hP2X3^{E111ANAP} (Fig. R8), slightly higher than that for the WT channel (0.78 ± 0.13 μM), the concentration of ATP (10 μM) used for VCF should be sufficient to activate most of this mutant channel.

The VCF experiment was quantified by measuring the shift of the peak emission wavelength (rather than the maximal fluorescence intensity), which provides a qualitative measure of ATP binding-induced conformational change at and near the substituted amino acid. Thus, unless the mutation dramatically disrupts ATP-binding to the receptor (being able to detect ATP-induced currents confirmed this not being the case here), small changes in ATP affinity would

not affect the interpretation of the VCF results as long as the ATP concentration used was relatively saturating.

Fig. R8. Concentration-response curves of ATP in $hP2X3E111ANAP$ to ATP. The solid line is a fit to the Hill 1 equation; $n = 3 - 5$

5. The shifts in emission λ_{max} values in Figure 2 are very modest. I think the authors need to do a control by adding 10 microMolar CTP or GTP (nucleotides that do not activate P2X3R) to verify that the emission changes are from specific binding of ATP and the conformational change in the receptor as opposed to just a change in emission because of increased charge (which might change the dielectric of the buffer..etc).

Response: Many thanks to the reviewer for bringing up this point, which is indeed one of the difficulties in using VCF in the mammalian expression system. This experiment was performed by detecting the emission spectrum shift of ANAP fluorescence as an indication of conformational changes, and is considered a method with high specificity for the labeled sites, low perturbation to the protein structure (Sakata, Jinno et al., PNAS, 2016), sensitive to minute conformations of receptors (Kalstrup and Blunck, PANS, 2013), etc. However, it also has the disadvantage of low fluorescence intensity. That is why emission spectra, instead of fluorescence intensities, before and after stimulation are used for comparison. The magnitude of the emission peak shift reported here is consistent with previous findings. For example, using the same method, Yang et al. detected shifts in the emission peaks of ANAP-incorporated TRPM8 channels in ranges of 2-4 nm and 4-8 nm (Xu et al., Nat Commun, 2020;

Yang et al., PANS, 2020); for the P2X2 receptor, Andriani et al. also reported shifted values that agreed with our results (Andriani et al., Nat Commun, 2020 and elife, 2021). Since the emission spectrum shift reflects a change in local environment, the degree of the emission peak shift differs depending not just on the type of receptor but also on the location of the specific amino acid in that receptor (Yang, et al., Nat Commun, 2018; Xu et al., Nat Commun, 2020; Sun et al., Science Bulletin, 2022). Using the same method, we have also detected large shifts in the TRPV1 receptor (Sun et al., Science Bulletin, 2022).

As pointed out by the reviewer, the possibility does exist that the negative charge of ATP could alter the solution properties and thereby affect the ANAP fluorescence. However, the fact that there were no significant changes in the emission spectrum in control cells that expressed the WT channel (new Fig. 2c,d, Note: these cells were also incubated with ANAP and contained intracellularly aggregated ANAP at the time of measurement) suggests that the application of ATP alone has minimal effect on ANAP fluorescence.

In addition, we have created a control ANAP-incorporated P2X3 receptor with the flUAA inserted at V143, a residue that is not expected to experience a change in local environment upon ATP activation since it is exposed directly to the solution and essentially has no surrounding amino acids to interact with (Fig. R9a). As with WT, there was no significant difference in the emission peak of V143^{ANAP} after ATP treatment ($P > 0.05$, paired t-test, $n = 5$, Fig. R9c). The changes in V143^{ANAP} were also not significantly different from the values of the changes in systemic "noise" (WT stands for systemic noise: fluorescence attenuation as well as non-specific intracellular aggregation of ANAP) ($P > 0.05$, unpaired t-test, $n = 5 - 6$, Fig. R9d), indicating that some positive insertions, such as E111^{ANAP}, are indeed associated with the corresponding sites of emission peak shifts in conformational changes (new Fig. 2c). This new data has been integrated into the revised manuscript as a control.

Fig. R9. ATP induces no change in the emission spectrum of V143^{ANAP}. (a) Location of V143 in the extracellular side of P2X3, exposed to the solution and outside of IP-HD. (b) Representative emission spectra of cells expressing V143^{ANAP} (black, bath solution; red, solution containing 10 μM ATP). (c-d) Summary of peak shifts in ANAP emission wavelengths in the absence and presence of ATP (n = 5-6, P > 0.05, ATP-treated vs. control, paired t-test (c) and P > 0.05 V143^{ANAP} vs. WT(d)).

The reviewer's suggestion on using "control by adding 10 μM of CTP or GTP to verify that the change in emission is from specific binding of ATP" is a very good one. Unfortunately, both CTP and GTP can also strongly activate the hP2X3 receptor (Fig. R10), although they exhibit 20 to 50-fold lower affinities than ATP in some other P2X receptors (Kasuya et al., Sci Rep, 2017).

Fig. R10. Representative current traces of P2X3 evoked by CTP (left) and GTP (right).

6. For the disulfide bond experiments shown in Figure 3, the authors do a nice job showing that oxidizing conditions limit ATP-induced current. They conclude that this is because the disulfide bond prevents the movements necessary for channel opening. However, an alternative explanation is that disulfide bond formation prevents ATP binding or decreases ATP affinity so the current is smaller at any given concentration of ATP. The authors need to show that the affinity for ATP

binding is unchanged when the disulfide bond is formed. Thus, ATP binding still occurs but the conformational change cannot occur because of the disulfide bond. I think the same thing can be said for the DTNB experiments and the zinc ion bridge experiments.

Response: We thank the reviewers for suggesting this point. As suggested by the reviewer, we measured the affinity of ATP on hP2X3^{L127C/T202C} before and after a disulfide bond formation and obtained EC₅₀ values of 1.40 ± 0.29 and 2.09 ± 1.01 μM , respectively (Fig. R11a). This indicates that the disulfide bond formation does not alter ATP affinity significantly. For Fig 3c, we used a near-saturating ATP concentration of 10 μM . The current magnitude changes caused by the disulfide bond formation and breakage, thus, represent most likely a gating effect rather than a change in ATP affinity.

While the engineered zinc bridge in hP2X3^{L127H/T202H} had almost no effect on the EC₅₀ of ATP (0.756 ± 0.323 μM and 0.784 ± 0.131 μM for hP2X3^{L127H/T202H} and hP2X3^{WT}, respectively, Fig. R11b); the EC₅₀ of hP2X3^{D158C} ATP with the covalent modification had a slight change (1.36 ± 0.07 μM , Fig. R11c). For hP2X3^{L127H/T202H} without Zn²⁺ and unmodified hP2X3^{D158C}, we used 1 μM ATP to induce currents. To test if these mutations alter their apparent affinity for ATP, we normalized the current evoked by 1 μM ATP to that by the saturating ATP concentration. The results (0.508 ± 0.065 vs. 0.545 ± 0.106 for P2X3^{L127H/T202H} vs. hP2X3^{WT}, $P > 0.05$, unpaired t-test; 0.508 ± 0.065 vs. 0.476 ± 0.106 hP2X3^{D158C} vs. hP2X3^{WT}, $P > 0.05$, unpaired t-test, Fig. R11d) suggest that 1 μM is about the EC₅₀ for both the WT and mutant channels. Thus, neither the mutation nor the Zn²⁺ bridge/covalent occupancy modification altered the apparent affinity of these mutants to ATP.

Fig. R11. Effects of disulfide bond, engineered zinc bridge, and DTNB modification on the ATP affinity of hP2X3 mutants. (a) Curve fitting of concentration-response to ATP of hP2X3^{L127C/T202C} under conditions of disulfide bond formation (H₂O₂) and breakage (DTT). (b) Concentration-responses to ATP of hP2X3^{WT} and hP2X3^{L127H/T202H} under the condition of zinc bridge formation. (c) Concentration-responses to ATP of hP2X3^{WT} and hP2X3^{D158C} treated with 2 mM DTNB. Normalized current is represented by I/I_{max} ; values are mean \pm SE, $n \geq 3$; Hill equation was used for fitting. (d) Ratio of current evoked by 1 μ M ATP to that by the saturating ATP concentration for hP2X3^{WT}, hP2X3^{L127H/T202H} and hP2X3^{D158C}, $P > 0.05$, unpaired t-test, $n = 3-5$.

To further illustrate the above point, we activated hP2X3 with the saturating ATP concentration (100 μ M). We found that Zn²⁺ still inhibited the ATP current of hP2X3^{L127H/T202H} (Fig. R12a), consistent with the results of original Fig. 3 (1 μ M ATP activation; also shown in Fig. R12b). This further demonstrates that the effect of Zn²⁺ is not due to a change in apparent affinity for ATP.

Fig. R12. Effect of Zn²⁺ ATP-evoked currents of WT hP2X3 receptors and its L127H/T202H mutant. Currents were induced by 100 μM (a) and 1 μM (b) ATP at intervals of 7 min. Data represent the ratio of ATP-evoked currents after Zn²⁺ treatment to that before treatment. Individual data points are shown as circles, n = 3 - 6, *P < 0.05 compared to WT channels, by unpaired t-test.

7. Did the authors ever show that Gefapixant also blocks the ATP-induced fluorescence changes in flUAA Anap? The authors state that incorporation of flUAA Anap into sites inside the IP-HD would “most likely interfere with PSFL2915 binding”. Were these experiments done? Does it actually interfere with PSFL2915 binding? If Gefapixant binds to a different site than PSFL2915 (outside the IP-HD), can gefapixant be used to block the emission shifts shown in Figure 2?

Response: We thank the reviewers for pointing this out and we have added the following experiments showing that gefapixant (200 nM) did not block the ATP-induced flUAA ANAP fluorescence change as PSFL2915 did (P > 0.05, paired t-test, n = 7, Fig. R13).

Fig. R13. Gefapixant does not interfere with ATP-induced flUAA ANAP fluorescence change. (a,b) Typical emission spectra of P2X3^{L297}ANAP before and after ATP stimulation in the absence (a) and presence of gefapixant (b). (c) Pooled data. Each line represents a pair of

measurements of ATP-induced ANAP emission peak shift in the same cell before and after addition of gefapixant (200 nM); n.s. $p > 0.05$, paired t-test; $n = 7$.

As suggested by the reviewer, we also analyzed the effect of flUAA Anap insertion into the internal site of IP-HD on PSFL2915 binding. Since the P2X3 receptor current is small after flUAA Anap insertion, we determined the inhibition of hP2X3^{L297ANAP} by PSFL2915 using the current density measurements. hP2X3^{L297ANAP} was still significantly inhibited by PSFL2915 (500 nM) ($P < 0.05$, unpaired t-test, $n = 6 - 8$, Fig. R14).

Fig. R14. Effect of PSFL2915 on the current density of hP2X3^{L297ANAP} induced by 10 μ M ATP. * $P < 0.05$, unpaired t-test, $n = 6-8$.

Reviewer #4 (Remarks to the Author).

The authors present a generally very well written and logically flowing study where they have used state-of-the-art molecular biology, cell biology, chemistry, biochemistry etc methods and also animal models. The topic is very interesting and significant in that it reports a natural compound quercetine and its new, more potent derivative as cough suppressants acting via the P2X3 receptor in an allosteric manner. The derivative PSFL2925 shows equipotent inhibition of the P2X3 cation channel as gefapixant, a commercial drug. However, quercetine and PSFL2925 do not show taste disturbing effects in animals like the commercial compound does. The figures are highly informative and generally of high quality. The language is very good. I recommend the paper to be published after minor revisions. I have provided my detailed comments and suggestions in the annotated manuscript pdf file.

Response: We are very appreciative of the kind words of the reviewer and have added additional experiments and revised the manuscript according to the reviewers' suggestions. We also appreciate the reviewer's very detailed comments on the full text, including sentence, wording, organization, and some other issues. Writing concerns have been adjusted in the revised manuscript (see blue text in the revised manuscript). Relevant questions and response to suggestions are answered/explained in detail below.

line 67. What is their mechanism of action/binding site?

Response: Binding sites for BAY-1817080 (Bayer), BLU-5739 (Bellows) and S-600918 (Shionogi) have not been reported. However, current unpublished data from our laboratory suggest that the binding site for S-600918 and its derivatives is in the outer region of the superior vestibule of hP2X3 (the manuscript is in revision in the BRITISH JOURNAL OF PHARMACOLOGY; Fig. R15). We have also previously demonstrated the druggable allosteric site formed by the left flipper and lower body domains for the binding of gefapixant (Wang J et al., PNAS, 2018). Thus, the currently known sites of action of these allosteric drugs are all different from that of quercetin and PSFL2915 (IP-HD). However, we cannot exclude BAY-1817080 and BLU-5739 from binding to IP-HD at this point, as their mechanism of action has not been disclosed.

Fig. R15. The action sites of DDTPA and S610098 are located between the β 2, 3 and 4 and β 13, 14. (a) Chemical structures of S-600918 and DDTPA. (b) Concentration-response curves of S-600918 for hP2X3 WT, hP2X3^{D79F}, hP2X3^{A283F}, hP2X3^{K284C}, hP2X3^{R295A}, hP2X3^{M96E}, hP2X3^{M96F}, hP2X3^{L298F}, hP2X3^{Q85A}, and hP2X3^{P83A}. (c) Superposition of the two interaction modes of DDTPA with an hP2X3 homotrimer derived from MetaD (orange) and *in silico* docking (green). Key residues are indicated with sticks for emphasis. Compared to *in silico* docking, MetaD also has a change in hP2X3 receptor conformation (because the site is composed of rigid β -sheets, there is only little conformational change). In order to show their differences more clearly, only the differences in the poses of small molecules obtained by the two computational approaches are highlighted.

Line 73. Pls refer here to the Figure where one can see these domains.

Response: A new **supplemental Figure 2** in the revised manuscript adds the labeling of each domain of the P2X3 receptor (also attached in **Fig. R16**).

Fig. R16. Domain organization of P2X3 receptor. (a) Cartoon representation of the "dolphin-like" single subunit of the hP2X3 structure. Each domain is labeled with a different color. (b) Amino acid sequence alignment of residues in the head, dorsal fin (DF), and left fin (LF) domains of the various P2X subtypes.

Line 85. Please check the order of the figures...should not S1 come before S2?

Response: We have reorganized the figures and their citations in the revised manuscript.

Line 117. Is this a result from MD simulations or just an observation when comparing the two structures, please stat this clearly...if an observation, the tempus could be present, not past: e.g. "based on the structural comparison of the resting and open states, the volume of...seems to undergo....."decreases from ...mainly results....

Response: We apologize for the ambiguity in the writing of the original manuscript, which was obtained by direct analysis of the resting and open states of hP2X3. We have improved the writing based on the reviewers' suggestions (lines 103 – 104, page 6).

Line 120. Is this the same as "resting"? Or is it the open state without ATP?

Response: This refers to the "resting" state. To avoid misunderstanding, we have changed the word "apo" to "resting" in the revised manuscript.

Line 128. The random conformational selection of a unique 'snapshot' structure...?

Response: We thank the reviewer for raising this point. We have modified in the revised manuscript. Protein structures obtained by structural determinations such as X-ray diffraction or cryo-electron microscopy are not necessarily the only conformations of the receptor in the physiological state. Because protein conformations are often in a dynamic process of change, the structure determined may be only one of the intermediate conformations, which we have referred to as a "snapshot". However, subsequent CMD simulations showed that the determined structure of hP2X3 remains in a relatively stable state.

Line 145. Why? what is the idea behind this method... what is CUA?

Response: In the assay that combines flUAA incorporation and VCF analysis to detect conformational changes, in order to distinguish from the 20 natural amino acids, the stop codon (TAG, 3 bases on the RNA chain for UAG, Fig. R17) is selected as the codon to guide the localized expression of ANAP in the target protein. CUA is the reverse complementary pairing with UAG engineered in the plasmid, and ANAP is then inserted into the corresponding protein sequence according to the base-pairing principle after binding to the tRNA with CUA base sequence by synthetase.

Fig. R17. Chemical structure of ANAP and strategy to incorporate L-Anap into the P2X3 receptor. Plasmids coding for Anap tRNA synthetase (AnapRS) and the corresponding tRNA, as well as a plasmid carrying the genes encoding the receptor, were transiently transfected into HEK293T cells. The cells were then cultured in ANAP-containing medium for at least 24 h to ensure that ANAP was integrated into the proper location of the receptor.

Line 151. How about 134? see next page...line 170

Response: The sites described here are all mutations on hP2X3 WT, while 134^{ANAP} is a mutation on hP2X3^{Y37A}, and therefore we describe it in line 170.

Line 213. What is the unit here?

Response: These values are ratios of currents before and after drug administration and therefore have no units.

Line 220. Rations of what? currents?

Response: Yes, it is the ratio of amplitudes.

Line 284. How did you choose these CV?

Response: We thank the reviewer for bringing up this important issue and we do apologize that in our original manuscript we incorrectly described the two collective variables (CVs) settings used during MetaD running. The correct CVs settings in the original 80 ns MetaD are shown below and have been corrected in the revised manuscript (lines 279-280, page 13). For ligand-receptor binding, we generally choose a distance CV: the key interaction of the receptor with the ligand, used to probe the ligand itself or some moiety of the ligand that requires enhanced conformational sampling, for its proximity and binding process with the receptor; another CV is the dihedral angle parameter of the ligand, which is in principle free to rotate and is used to probe the best conformation for the proximity and binding process of the ligand with the receptor. Since the aromatic ring formed by quercetin C1'-C6' is inserted below loop formed by residues 108-118 (new Fig. 1a), the cavity is small and conformational sampling is limited. Whereas the heterocycle formed by C1-C10 faces the huge ATP-binding site jaw (new Fig. 1b) and the cavity below the loop formed by residues 122-134, enhanced conformational sampling is required here. Thus, the O1-C2-C1'-C2' chain forms a dihedral angle $\chi_{C2'-C1'-C2-O1}$ (new supplementary Fig. 7e) that is in principle freely rotatable from -180° to $+180^\circ$ (Fig. R18), we used this as the CV1. In addition, we took as CV2 the distance between the quercetin C8-phenolic hydroxyl group and the hydroxyl group of Y114 in the space below the loop formed by residues 122-134 (Fig. R18), and the distance wall was set to 20 Å.

Fig. R18. Collective variable (CV) settings during MetaD simulations. The thick yellow, orange, green and sky-blue dashed lines indicate hydrogen bonding, proximity interactions, cation- π and π - π stacking interactions, respectively. The thin purple dashed lines represent ionic interactions.

Line 288. This was not mutated? see below.

Response: Sorry for the missing mutation Y114A in the original manuscript. It is now integrated into the revised version of new Fig. 6f.

Line 293. S110 was not listed above?

Response: Although no direct interaction of S110 with quercetin was shown in the CMD results, the site is near the top of the binding pocket. Therefore, the conformation of IP-HD was disrupted by mutating the Ser to a bulkier side chain with Phe to verify our hypothesis (Fig. R19).

Fig. R19. Locations of S110 (a) and F110 (b) in the IP-HD of the hP2X3 WT and hP2X3^{S110F}.

Line 303. Do you have any idea why so? Is there some allosteric effect?

Response: Yes, as the reviewer speculated, this is an indirect conformational change. Because K284 is closer to some amino acids that make up IP-HD, the K284R mutant may have changed the conformation of IP-HD, thus weakening quercetin binding (**Fig. R20**).

Fig. R20. Location of K284 around the IP-HD of P2X3.

Line 307. Why did you choose this residue?

Response: We thank the reviewer for raising this point. Because some key amino acids are in the flexible loop structure (such as E156, E158, K113 and Y114), if covalent occupancy is introduced by targeting Cys in the loop structure, it may not have much effect on the IP-HD conformation. Secondly, if the inhibition of quercetin after mutation of a particular key amino acid changes so much that it cannot even be bound, then a further comparison before and after covalent occupancy is not very meaningful. In contrast, S67 is located at a relatively rigid β -fold in the lower IP-HD (**Fig. R21**), and the change of quercetin inhibition before and after covalent occupancy would be more obviously, we thus chose this site for covalent modification.

Fig. R21. Location of S67 in IP-HD.

Line 398. Why did you use a smaller amount here? Did you also try with the full dose?

Response: We thank the reviewer for bringing up this point. PSFL2915 and gefapixant had similar cough suppressant effects at the same dose. While gefapixant still exhibited side effects affecting taste at lower doses, the higher dose of PSFL2915 exhibited no side effects, which is more indicative of PSFL2915's ability to avoid side effects.

As also suggested by reviewer #2, we tested the side effects of two higher doses of gefapixant (10 and 20 mg/kg) on taste perception of rats. The results showed that the increased doses had greater side effects on taste perception in the animals (**Fig. R22**).

Fig. R22. Effects of gefapixant, quercetin, and PSFL2915 on taste perception in rats. n.s. $P > 0.05$, * $P < 0.05$ compared with vehicle; paired t-test; $n = 10$.

Line 427. Pls check the use of terms.

Response: We have corrected this in the revised manuscript.

Line 436. How high dose of quercetin can/should be used? Maybe there are other side effects with higher doses?

Response: We thank the reviewer for suggesting this point. Generally, a dose of 0.5-1 g/day is a safe dose (Heinz, Henson et al. 2010). This clinical trial study showed that 5 g/day of quercetin (almost the highest dose that could be taken in a clinical trial) had no significant side effects (Lu, Crespi et al. 2016), demonstrating the safety of quercetin.

Line 520. normal mode analysis? this was not mentioned in the study...

Response: We are very sorry for this typo. We have removed it from the revised manuscript.

Line 541. What is RS?

Response: RS is an abbreviation for tRNA synthetase. In VCF experiments, plasmids encoding tRNA and Anap tRNA synthetase (AnapRS, see also Fig. R17) need to be constructed to establish the translational expression system of ANAP after co-transfection.

Line 586. Where is the Kd seen in the figure?

Response: The kD in the figure refers to the molecular weight unit of the protein. Kd in the figure legend refers to the dissociation constant (Kd) obtained by fitting in the MST binding assay. To avoid unnecessary confusion, in the modified figure legends, we spell out “dissociation constants” directly instead of using the abbreviations.

Line 635. With vehicle?

Response: The vehicle is 0.5% CMC-Na solution.

Line 662. Why are the some of the residue labels in green? Please define.

Response: The green font represents the corresponding residues of the receptor that interacts with the Mg²⁺. We have updated the information in the revised manuscript.

Line 665. Is this for the open state?

Response: We are very sorry for the unnecessary misleading caused by typos and writing ATP as quercetin. Yes, this is the ATP-bound P2X receptor that is in the open state. We have corrected it in the revised manuscript.

Line 674. Is it possible to make similar (comparable) figures? The view points seem to be so different that it is difficult to compare them...

Response: Thanks to the reviewer's suggestion, and we have made a figure in new Fig. S2 of the revised manuscript.

Line682. Selected virtual hit compounds? Or already the experimentally tested hits?

Response: Yes, this is the structure of the compound that has been electrophysiologically tested.

Line682. Why is quercetin in gray? why did you not consider also myricetin as it seems to inhibit as much as quercetin?

Response: In our original manuscript, quercetin was represented in gray to distinguish it from the key compounds and other compounds in the subsequent study. To avoid unnecessary

confusion, we changed the color to brick red in the new Fig. S3. Since quercetin is used to relieve respiratory diseases such as cough, we chose quercetin for further study.

Line 702. ionic interactions are with negatively charged amino acids, which cannot directly interact with quercetin as it does not have any positive functional group...please make it clear that the interactions is with/through the Mg²⁺ cation...

Response: We thank the reviewer for raising this point. In fact, as the reviewer speculated, it is DESMOND's simulated interaction diagram analysis module that treats Mg²⁺-assisted/bridge ionic interactions as ionic interactions, which is different from the direct positive and negative charge interactions between ligands and receptors that we usually consider (salt bridges or ionic interactions). We have made an additional illustration in the new Fig. S7 legends.

Line 712. What is the value range on the y-axis?

Response: We have added the value range to the new Fig. S7 y-axis, which represents the sum of the corresponding simulated time-length receptor-ligand interactions, including hydrogen bonding, salt bonding, hydrophobic interactions, etc.

Line 715. Contacts by the particular residue?

Response: Yes, this is an interaction with a specific amino acid. The darker the orange color, the more number of contacts the residue has with the ligand, as some amino acids have multiple specific contacts with the ligand

Line 717. Pls define the red dashed interaction lines.

Response: They represents some important polar contacts, including Mg²⁺-assisted ionic interactions, hydrogen bonding, etc. We have made additional illustrations in the new legend of Fig. S7.

Line 791. This was not very clearly described in the results? Is this a previous result that has been published already?

Response: We are very sorry that we did not describe it clearly in our original manuscript and we have rephrased it in the revised manuscript. This is the result of our first screening. The manuscript does not contain any data that have been published in other journals.

Line 792. Did you use many structures? Please give then the PDB IDs for all. Please add also their resolution and the reference to the articles that report the structures.

Response: No, we chose only one structure because human P2X3 (PDB ID: 6AH4) is a full-length structure with good sequence identity to P2X2. We did not need to use the truncated zfP2X4 or the more sequence-diverse rP2X7 structures as multiple templates for homology modeling. Information about their resolution and references has been added in the revised manuscript (line 810, page 48).

Line 793. This reference is wrong...Caver is not Modeller...Sali lab has created the Modeller tool.

Response: We are very sorry about this and have corrected it in the revised manuscript.

How many models were created, what was the modeling alignment (give for example in the Supplementary data?), which of the models did you choose for further investigations and what was the criteria? Stereochemical quality Any comment on the quality in the results?

Response: We built 30 models. For information about sequence alignment, we used information from Nature papers in 2009, 2016 and 2012 (Kawate, Michel et al., Nature, 2009; Hattori and Gouaux, Nature, 2012; Mansoor, Lu et al., Nature, 2016). Because the 3D architecture of the P2X receptor family members is very conserved (the structure of the moonflower tick P2X receptor (amP2X) is almost identical to that of hP2X3 (Kasuya, Fujiwara et al., Cell Rep, 2016)); also, the 3D architecture of multiple structures of P2X3, 4 and 7 that have been resolved are almost identical (Kawate, Michel et al., Nature, 2009; Mansoor, Lu et al., Nature, 2016; Karasawa and Kawate, Elife, 2016; Kasuya, Yamaura et al., Nat Commun, 2017), the structures are essentially treated as identical for the various subtypes of the P2X family. Because there are too many supplemental figures for this manuscript and detailed sequence comparison is provided in the Nature paper in 2009 (Kawate, Michel et al., Nature, 2009), we have not integrated the relevant information into the supplemental figure here.

Among these 30 output models, we prefer to calculate the DOPE score for each structure according to the MODELER Discrete Optimized Protein Energy (DOPE) method; and rank the models according to the DOPE score, and then select the model with the highest score. Meanwhile, the proportion of unreasonable amino acids in the selected model is tested with Procheck so that most amino acid conformations are distributed within a reasonable range as much as possible (if some amino acids in the template structure are already in the unreasonable region, we will keep the experimentally determined side chains of the structure). The model is then optimized (hydrogen addition, charge addition and hydrogen atom optimization) using Schrödinger's preparation wizard module; after local restraint optimization, no significant interatomic clashes (including between hydrogen atoms) occur in the homologous model. If collisions are still present in the restrained optimization, another model should be selected or the homologous model should be reconstructed. The optimized model is then carefully examined by hand for a final round of checks, and if there are positions that clearly contradict the commonality of the P2X structure, they are also excluded. After these steps, a usable model is obtained. Since the 3D structure of the P2X receptor family is

conserved, homology models of various P2X subtypes are trustworthy and can be used for a range of subsequent studies.

Line 794. Simulations were carried out using DESMOND (ref), as implemented in Maestro molecular modeling suite (Schrödinger, LLC) (ref). The simulation systems were constructed employing Maestro's System Builder function....Please see Schrödinger website for how to cite their software....software versions should also be given, which version of Maestro was used...etc. Protein preparation wizard/workflow was also apparently used to add the hydrogens etc...?

Response: Thanks to the reviewer for such careful reminding. We have modified references (lines 813-814, pages 48-49) as described on Schrödinger's website. For the version, we used Schrödinger Suite 2018. Maestro and Protein Preparation Wizard/Workflow also corresponds to Schrödinger 2018-1. The protein preparation wizard of Schrödinger Suite 2018 is very comprehensive and includes filling of missing side chains, hydrogen addition, charge addition, H-bond assignment, optimization of H-assignment and restrained optimization, etc.

Line 798 . The Orientations of Proteins in Membranes (OPM) database (pls give the ref).

Response: We have added references in the revised manuscript.

Line 800 Na⁺ counter ions?

Response: Yes, if the net charge of the system is negative, we add the Na⁺ counterion; if it is positively charged, we add Cl⁻.

Line 801. why?

Response: Because 150 mM is the extracellular NaCl concentration of most neurons or excitable cells under physiological conditions and the NaCl concentration at which we performed electrophysiological experiments, the concentration we chose was 150 mM.

Line 805. References for all methods should be added; Line 807. Ref; Line 813. Ref.

Line 811. Reference should be given; please make sure you did not use the U-series instead as this is the default in the more recent versions of Maestro/Desmond...

Response: We have added relevant references or URLs. The Schrödinger suite is version 2018-1, and Desmond is version 2018-4 free for academic users.

Line 815. Did you also study the PSFL2915, or only quercetin? would be interesting to know how the sulfogroups interact...

Response: As suggested by the reviewer, we added 1.2 μ s of CMD simulations of the PSFL2915/hP2X3 complex. PSFL2915 still dominated the Mg^{2+} -mediated interaction with E111 and E156, while its sulfogroups has greater interactions with R281 and R65 at the periphery of IP-HD early in the simulation (~0-30 ns) (Fig. R23 a, d), and in the subsequent 1000 ns, the interaction with R299 and R65 took over (Fig. R23 a, d). However, it is noteworthy that both sulfogroups remain conformationally unstable throughout the simulation (Fig. R23b - purple and lime-green rotatable bonds; Fig. R23c, the ligand RMSD also fluctuated considerably). At the same time, we demonstrate that PSFL2915 acts on IP-HD (S67C-DTNB covalent occupation still greatly attenuates the action of PSFL2915) rather than competing with ATP on the binding site (new Fig. 8a, b). Thus, we suggest that the phenolic hydroxyl group of PSFL2915 interacts with IP-HD with the aid of Mg^{2+} , while its sulfogroups interact with amino acids within IP-HD or with Lys and Arg in the periphery of hP2X3 IP-HD (Mg^{2+} may be directly and indirectly involved). The above two polar interactions together increase the binding of PSFL2915 to IP-HD. More exact interactions still need to be determined by structure determination of the PSFL2915/hP2X3 complex.

Fig. R23. CMD simulations to study the interaction of hP2X3 with PSFL2915 at the atomic level. (a) The interaction between P2X3 and PSFL2915 was monitored throughout the MD simulations. hP2X3/PSFL2915 interactions were classified into four types: hydrogen bonds (green), hydrophobic (light purple), ionic (pink), and water bridges (blue). The stacked histograms are normalized over the course of the trajectory. (b) Backbone root-mean-square deviation (RMSD) analysis of the binding process of PSFL2915 to histograms. (c) Torsion plots of PSFL2915 summarizing the conformational evolution of each rotatable bond every 10 ps throughout the simulated trajectory. The radial plots represent the conformation of the torsion bodies. Center of the radial plot represents the beginning of the simulation, plotting the temporal evolution in the radial direction outward. The histogram summarizes the data of the corresponding radial plot, which represents the probability density of the torsion. The relationship between histogram and torsional potential gives insight into the conformational strain that the ligand underwent to maintain the hP2X3-bound conformation. (d) Time-

dependent contacts between P2X3 and PSFL2915. The upper panel represents the total contacts of P2X3 with PSFL2915; the lower panel represents the contacts of individual amino acids with PSFL2915. The darker the orange color, the more number of contacts the residue has with the ligand, as some amino acids have multiple specific contracts with the ligand.

Line 819. which spacing?

Response: Sorry for the confusion caused by the term 'spacing'. The specific parameters of the MetaD simulation are as follows: 1. width of the repulsive Gaussian potential (root mean square width) in angstrom or degrees. The distance is 0.05\AA , the angle is 2.5° , and the dihedral is 5° . 2. the height of the repulsive Gaussian potential (kcal/mol). 3. the time interval for adding the repulsive Gaussian potential. Therefore, the term "spacing" in the original manuscript refers to the interval for adding the repulsive Gaussian potential (0.12 ps). We have made changes in the revised manuscript.

Line 820. How do you observe this, what is the parameter to be followed?

Response: As mentioned above regarding the CV setup (Fig. R18), we set two collective variables: 1. one is the two-atom distance parameter for the key new interaction of the ligand and the receptor, while setting the 'wall' distance (20\AA); 2. the other is the dihedral angle parameter, ranging from -180° to $+180^\circ$. When the distance inside the wall (20\AA) and the angle from -180° to $+180^\circ$ has been fully evaluated, these two CVs are considered as free

diffusion during the repulsive Gaussian potential (Fig. R24).

Fig. R24. An example showing the convergence of metadynamics run. (a) Free-energy profile versus the dihedral angle $\chi_{C2'-C1'-C2-O1}$ of quercetin and the O...O distance between the quercetin C8-phenolic hydroxyl group and the hydroxyl group of Y114. The black dot denotes snapshots from metadynamics run with a free diffusivity along the defined CVs. (b, c) Time evolution of two collective variables (CV1, b; CV2, c) during a metadynamics run. Data were plotted as a scatter graph rather than a line graph for clarity.

Line 821. Please give the correct reference for the tool.

Response: We have removed the incorrect reference.

Line 824. This is likely not the original article describing the method? Please cite (also) the original?

Response: We have added the original reference.

Line 826. Pls check.... something is missing, and should be deleted 'd'?

Response: "d CVs" should be "d CVs". CVs are our own selected collective variables, and "d CVs" is the set of all CVs in the metadynamics runs.

Line 828. Transient what?

Response: Means transient conformation/state.

Line 834. Pls describe shortly how you screened, using Glide? Which mode? what was the criteria to select the hits for testing?

Response: Thanks to the reviewer's suggestion. In general, docking with Glide HTVS or SP is to avoid false negatives and docking with Glide XP is to avoid false positive results. A virtual compound library with millions of compounds could be screened by HTVS-SP-XP sequences, but since we screened here only a small library of 20-30,000 compounds (natural products and active compounds), the XP mode was used directly. We screened a small library of compounds to verify that IP-HD is an allosteric pocket that can be regulated by the active molecule. In subsequent studies, we can screen a library of compounds in the millions or more, such as Enamine and CHEMDIV, by the HTVS-SP-XP protocol to obtain novel P2X3 inhibitors with novel backbones.

REVIEWERS' COMMENTS

Reviewer #1 (Remarks to the Author):

I have reviewed a revised and improved version of the manuscript by C-R Guo and Collaborators on the structure and allosteric modulation of the human P2X3 channel by Quercetin and PSFL2915.

The authors have satisfactorily answered all my question. They have added additional controls for the VCF experiments and increased the number of observations.

I have no further questions.

Reviewer #2 (Remarks to the Author):

I want to congratulate the authors for an outstanding manuscript and for a very thorough rebuttal of the reviews received previously. This is a very well done and impressive body of research. It will be interesting to see if this therapeutic approach gains traction in patients.

Reviewer #3 (Remarks to the Author):

The authors have gone out of their way to address nearly all of the concerns of the reviewers.

I commend the authors on their thorough systematic responses and clear improvements to the paper.

I believe this manuscript is ready for publication!

Reviewer #4 (Remarks to the Author):

The authors have answered to all my concerns satisfactorily and in some cases even very thoroughly. One mistake I still found on page 50, line 858: ..."XP model". This should be "Glide extra precision (XP) mode". So mode, not model.

Point-to-point responses to reviewers' comments (*comments in italics*)

Reviewer #1 (Remarks to the Author):

I have reviewed a revised and improved version of the manuscript by C-R Guo and Collaborators on the structure and allosteric modulation of the human P2X3 channel by Quercetin and PSFL2915. The authors have satisfactorily answered all my question. They have added additional controls for the VCF experiments and increased the number of observations. I have no further questions.

Response: We are very grateful to the reviewer for his/her positive comments on the manuscript.

Reviewer #2 (Remarks to the Author):

I want to congratulate the authors for an outstanding manuscript and for a very thorough rebuttal of the reviews received previously. This is a very well done and impressive body of research. It will be interesting to see if this therapeutic approach gains traction in patients.

Response: We are very grateful to the reviewer for his/her positive comments on the manuscript.

Reviewer #3 (Remarks to the Author):

The authors have gone out of their way to address nearly all of the concerns of the reviewers. I commend the authors on their thorough systematic responses and clear improvements to the paper. I believe this manuscript is ready for publication!

Response: We are very grateful to the reviewer for his/her positive comments on the

manuscript.

Reviewer #4 (Remarks to the Author):

The authors have answered to all my concerns satisfactorily and in some cases even very thoroughly.

One mistake I still found on page 50, line 858: ... "XP model". This should be "Glide extra precision

(XP) mode". So mode, not model.

Response: We are very grateful to the reviewer for his/her positive comments on the manuscript. We have corrected this mistake in the revised manuscript.